# Recurring adaptive introgression of a supergene variant that determines social organization

Eckart Stolle [1,2,7✉], Rodrigo Pracana [1,7✉], Federico López-Osorio [1], Marian K. Priebe[1],
Gabriel Luis Hernández[1], Claudia Castillo-Carrillo[1], Maria Cristina Arias[3], Carolina Ivon Paris[4],
Martin Bollazzi [5], Anurag Priyam[1] & Yannick Wurm [1,6✉]

Introgression has been proposed as an essential source of adaptive genetic variation. However, a key barrier to adaptive introgression is that recombination can break down combinations of alleles that underpin many traits. This barrier might be overcome in supergene regions, where suppressed recombination leads to joint inheritance across many loci. Here, we study the evolution of a large supergene region that determines a major social and ecological trait in *Solenopsis* fire ants: whether colonies have one queen or multiple queens. Using coalescent-based phylogenies built from the genomes of 365 haploid fire ant males, we show that the supergene variant responsible for multiple-queen colonies evolved in one species and repeatedly spread to other species through introgressive hybridization. This finding highlights how supergene architecture can enable a complex adaptive phenotype to recurrently permeate species boundaries.

[1] School of Biological and Behavioural Sciences, Queen Mary University of London, London, UK. [2] Leibniz Institute for the Analysis of Biodiversity Change (LIB), Zoological Research Museum Alexander Koenig, Bonn, Germany. [3] Departamento de Genética e Biologia Evolutiva, Instituto de Biociências, Universidade de São Paulo, São Paulo, Brazil. [4] Departamento Ecología, Genética y Evolución, Facultad de Ciencias Exactas y Naturales, Universidad de Buenos Aires, Buenos Aires, Argentina. [5] Entomología, Departamento de Protección Vegetal, Facultad de Agronomía, Universidad de la República, Montevideo, Uruguay. [6] Alan Turing Institute, London, UK. [7] These authors contributed equally: Eckart Stolle, Rodrigo Pracana. ✉email: e.stolle@leibniz-zfmk.de; rodrigopracana@gmail.com; y.wurm@qmul.ac.uk

Alleles can be transferred between species by introgressive hybridization[1,2]. Is it possible for a complex trait involving alleles at many loci to evolve in one species and then introgress into another? One of the main barriers to this type of introgression is recombination, which breaks up stretches of DNA, producing haplotypes that combine alleles from donor and recipient species[3]. Genomic inversions can impede recombination to create supergene variants whereby entire genomic regions are inherited as units[4,5]. Recent studies have shown that supergene variants can spread, unhindered by recombination, not only among populations but also across species barriers[6–10].

In this study, we test competing hypotheses regarding the origin and evolutionary history of a supergene that determines whether colonies of the fire ant *Solenopsis invicta* have one queen or multiple queens. Several behavioral, morphological, and physiological traits co-vary with queen number. For example, queens from single-queen colonies are better at dispersing to new habitats, while multiple-queen colonies are more competitive at high population densities[11]. The queen number social dimorphism is controlled by two variants of a supergene region, SB and Sb, on "social" chromosome 16[12,13]. In single-queen colonies, all individuals carry exclusively the SB variant, and workers kill any potential additional queens. In multiple-queen colonies, many workers carry the Sb variant, but kill queens that lack the Sb variant[13]. The supergene region contains more than 470 protein-coding genes, most of which are present in both supergene variants[12]. The duplication or absence of some genes in Sb[14,15], and the divergence in sequence and expression of several others suggest that they contribute to the social phenotype[16]. In addition, Sb differs from SB by three large inversions, and Sb has expanded through the accumulation of repetitive elements[17,18] (estimated sizes are 20.9 Mb for SB and 27.5 Mb for Sb).

Interestingly, close relatives of *S. invicta* are also socially dimorphic and carry the same social supergene system[17–19]. Although this supergene system could be a trans-species polymorphism that evolved in the common ancestor of these species[17–19], or even a set of polymorphisms that originated independently in each species, it is possible that its spread[17,18,20] was instead mediated by introgressive hybridization. Here, using coalescent-based phylogenies, we show that the supergene system originated in *S. invicta* and spread to different species through the introgression of the Sb variant of the supergene.

## Results

**Coalescent construction of a species phylogeny and a supergene phylogeny.** To understand the evolutionary history of the social supergene, we compared a phylogenetic tree representing the relationships among *Solenopsis* species with a phylogenetic tree representing the history of the supergene region (Fig. 1a–d illustrate potential scenarios). For this, we first obtained whole-genome sequences of 368 samples of *Solenopsis*. These include 261 samples published by ourselves and others[12,14,16–18,21,22] and 107 samples that we additionally collected to expand taxonomic and geographical coverage (Supplementary Data 1 and Supplementary Fig. 1). Most of the samples are haploid males (365 out of 368). Using these genomes, we constructed a phylogenetic tree from each of 1728 informative single-copy genes[23] (Supplementary Data 2). We then applied a coalescent-based method[24] to reconstruct a species tree from 1631 single-copy genes mapping to chromosomes 1–15 and a supergene tree from 97 single-copy genes mapping to the supergene region of the social chromosome (Fig. 1e). In both trees, nodes critical for interpreting the histories of the species and the supergene are highly supported[24] (bootstrap support 98–100%, local posterior probabilities 0.89–1.00).

The species tree (left side of Fig. 1e) shows that males from the socially dimorphic species *S. invicta*, *S. macdonaghi* and *S. richteri* are organized into two sister clades. One clade includes all *S. richteri* males. The second clade includes *S. invicta* and *S. macdonaghi* males in a paraphyletic relationship, suggesting that this clade may represent a single species. We hereafter refer to it as *S. invicta/macdonaghi*. All samples from other species, representing *S. geminata, S. pusillignis, S. saevissima, S. interrupta, S. megergates* and the undescribed species *S. AdRX*[18,25], are organized into five clades branching off the base of the species tree.

**The social supergene variant Sb evolved in one species and subsequently introgressed into others.** Comparing the species tree with the supergene tree (Fig. 1e) allows us to infer the evolutionary history of the supergene. For males carrying the SB variant of the supergene and all but five males from other species (see below), the general topologies of the supergene tree and the species tree are congruent. However, all Sb males group into one clade that is sister to the SB *S. invicta/macdonaghi* clade. We can thus reject the hypothesis that the supergene is a trans-species polymorphism that originated before the speciation of *S. richteri* and *S. invicta/macdonaghi*. Indeed, if that were the case, each species would form one monophyletic group within the SB clade and another within the Sb clade (scenario in Fig. 1b). We can also reject the hypothesis that there were multiple origins of the supergene, whereby the *S. richteri* Sb samples would form a distinct sister group to the *S. richteri* SB samples (scenario in Fig. 1c). Instead, the grouping of Sb males as sister to the *S. invicta/macdonaghi* SB clade shows that the Sb supergene variant originated in chromosome 16 of *S. invicta/macdonaghi* after the divergence between this species and *S. richteri*. The presence of Sb sequences from other species within the *S. invicta/macdonaghi* Sb clade shows that Sb introgressed from *S. invicta/macdonaghi* into the other species (scenario in Fig. 1d). In addition, the hypothesis that the supergene arose through hybridization of two species with opposing orientations[6] of the supergene-equivalent region on chromosome 16 is unparsimonious, as it would have required the introgression of Sb back into *S. invicta/macdonaghi* soon after its origin. Instead, the differentiation between Sb and SB suggests that the inversions that suppress recombination between Sb and SB[12,17,18] in *S. invicta/macdonaghi* are conserved across species.

**Recurring ancient and recent introgression of the supergene into different species.** The Sb males from species other than *S. invicta/macdonaghi* are positioned in six Sb subclades within the supergene tree, suggesting that six independent introgression events occurred. The presence of two groups of *S. richteri* Sb samples within the Sb subtree (numbers 1 and 2 in Fig. 1e) could indicate that Sb introgressed twice from *S. invicta/macdonaghi* into *S. richteri* (see Discussion below). Strikingly, the other four cases of Sb introgression (numbered 3–6 in Fig. 1e) involve five Sb males positioned in early diverging clades of the species tree: one case involving *S. interrupta* (numbered 4; with two males), two cases involving *S. megergates* (numbered 3 and 5), and one case involving *S. AdRX* (numbered 6). All Sb haplotypes ultimately originate from the *S. invicta/macdonaghi* clade. However, in the supergene tree, one *S. megergates* Sb male (numbered 5) is adjacent to one of the *S. richteri* Sb clades (numbered 2), raising the possibility that Sb introgressed from one of these two species to the other secondarily.

A divergence time inference (Supplementary Note 1) suggests that the separation of SB and Sb occurred 0.97 million years ago (mya), after the split between *S. richteri* and *S. invicta/*

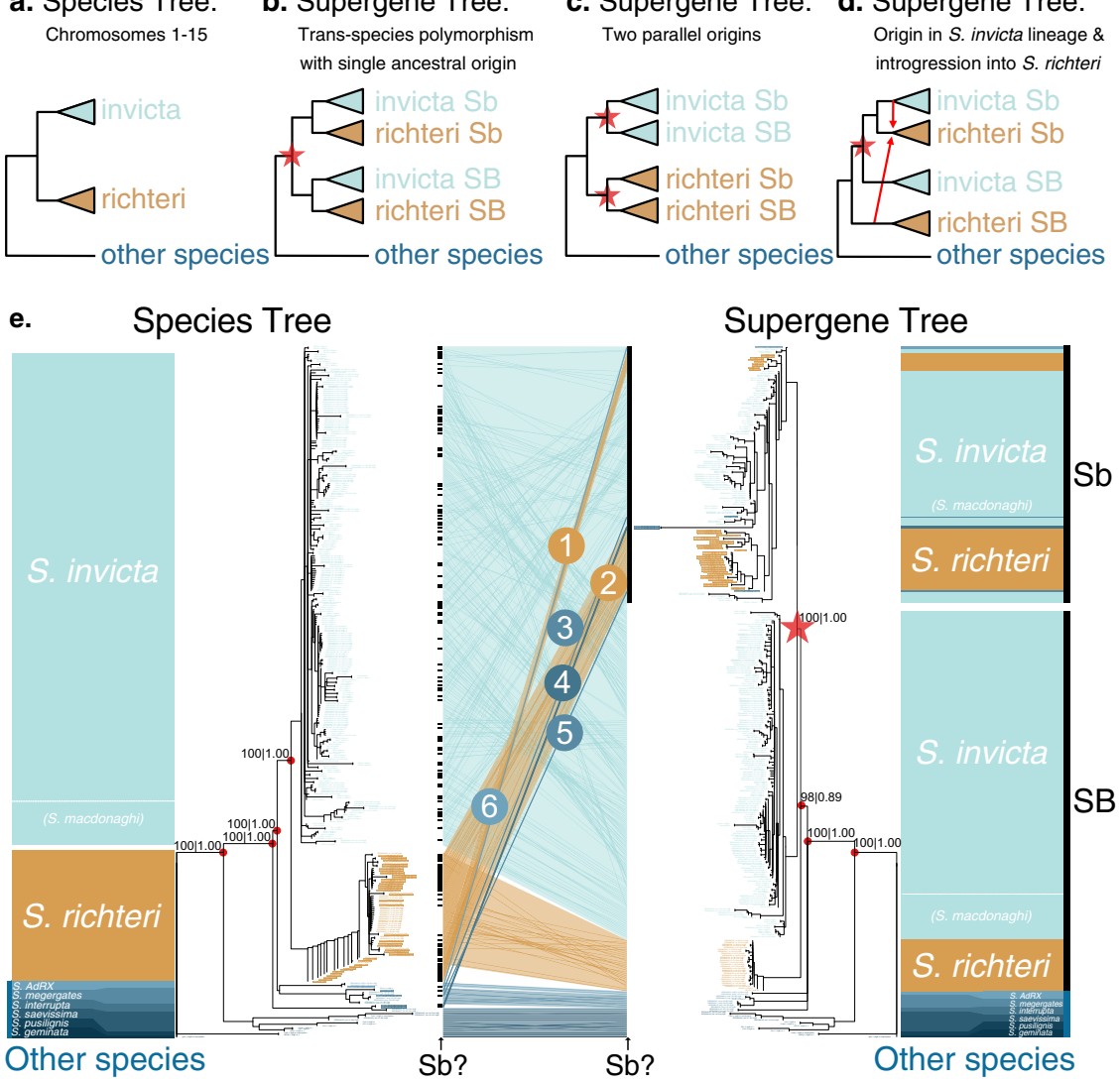

**Fig. 1 Hypothetical and empirical species and supergene phylogenetic trees. a–d** A simplified species tree (**a**), and hypothetical scenarios for the evolutionary history of the supergene (**b–d**). **b** SB and Sb supergene variants diverged (star) in the common ancestor of *S. invicta* and *S. richteri*; the supergene is thus a trans-species polymorphism. **c** Sb evolved twice from SB, representing independent origins (stars) after the separation of the two species. **d** Sb diverged in *S. invicta* and spread to *S. richteri* through introgression (arrows). **e** Empirical coalescent-based trees of 368 *Solenopsis* samples based on 1631 single-gene trees from chromosomes 1–15 (left; species tree) and 97 single-gene trees from the supergene region of chromosome 16 (right; supergene tree). Branches shorter than 0.05 were collapsed into polytomies. A tanglegram (middle) indicates the relative positions of each sample in both trees. Circled numbers highlight patterns consistent with introgression of Sb from *S. invicta/macdonaghi* into other species. On either side of the tanglegram black bars indicate where samples with the Sb genotype can be found in the two trees. Support values (ASTRAL bootstrap support | local posterior probability) are provided for key nodes of speciation and supergene differentiation.

*macdonaghi* (1.01 mya) and before the diversification of *S. invicta/macdonaghi* (0.81 mya) (Supplementary Figs. 2–4). All but one of the introgression events of the Sb supergene variant from *S. invicta/macdonaghi* into the other species occurred in the first quarter of its existence. The remaining introgression event is one of the two introgressions into *S. megergates* (numbered 3 in Fig. 1e); the age of the node representing this introgression event is within the age range of nodes separating individuals of the same species (Supplementary Note 1). Consistent with this, the putatively more ancient *S. richteri* Sb clades include long branch lengths in the supergene tree (clades 1 and 2 of Fig. 1e), and high nucleotide diversities (Supplementary Fig. 5 and Supplementary Note 2). In contrast, the younger Sb clade of *S. megergates* has a highly similar haplotype to those found in three

*S. invicta/macdonaghi* Sb males we collected within 5.2 km of each other (Supplementary Note 3, Supplementary Fig. 6, and Supplementary Data 3). Altogether, our results suggest that introgression of the Sb supergene variant is a common mechanism for rapid adaptive evolution among fire ants and that this mechanism is not restricted to a particular Sb source haplotype or recipient species.

**Discussion**

Our findings contradict previous suggestions that the Sb supergene variant evolved in a common ancestor of *S. invicta* and *S. richteri*[18] or within *S. richteri*[20]. By incorporating twice as many samples as the most recent large-scale study[18], our work adds clades that had not yet been sampled. We focused on single-

copy genes, which allowed us to avoid complex genealogies and potential genotyping artifacts caused by read mis-mapping of copy-number variants of genes and repetitive elements; such variation is particularly common in the degenerating Sb variant of the social chromosome[14,17,21,26]. Finally, rather than implying that all sequences share one evolutionary history, our coalescent-based tree reconstruction approach should be robust to gene tree discordance due to tree reconstruction errors and processes including incomplete lineage sorting and gene conversion[24,27].

Nevertheless, we pursued five additional lines of analysis to challenge our finding of introgression. First, we produced a supergene tree and a species tree following the concatenated alignment approach similar to another recent study[18], but after having excluded ambiguous regions of the genome (Supplementary Note 4 and Supplementary Fig. 7). The resulting phylogenies are consistent with those from the coalescent-based tree reconstructions (Fig. 1e) and not the previous studies, supporting the introgression of Sb from *S. invicta/macdonaghi* into each of the four *Solenopsis* species mentioned above. A notable difference between our two approaches is that the Sb samples of *S. richteri* named as group 1 in the coalescent-based tree are placed into one monophyletic group with the remaining *S. richteri* Sb samples in the concatenation-based tree, suggesting that Sb introgressed only once from *S. invicta* into this species. In addition, the Sb sample from *S. AdRX* (group 6 in the coalescent-based tree, Fig. 1e) is placed as a sister taxon to a relatively basal clade of Sb in the concatenation-based tree, suggesting that the introgression into *S. AdRX* occurred relatively early. Second, we created de novo genome assemblies using a subset of higher-coverage samples and derived an independent dataset of 2161 conserved single-copy ortholog sequences (Supplementary Note 1 and Supplementary Data 4). Phylogenetic inference yielded a dated species tree (Supplementary Fig. 2) with a topology that was fully consistent with the species tree from the coalescent-based tree reconstruction (Fig. 1e). Third, we quantified the support for introgression relative to incomplete lineage sorting using branch length distributions (QuIBL[7]; Supplementary Note 5) of trees built from single-copy genes from the supergene region. This analysis also supported the introgression of Sb between *S. invicta/macdonaghi* and the other four species (Supplementary Figs. 8–11). Fourth, if Sb originated in *S. invicta/macdonaghi*, any Sb haplotype should have more alleles in common with the SB variant of *S. invicta/macdonaghi* than with the SB variants of the other species. Indeed, in an analysis with equal number of samples for each species and supergene variant, we found that the Sb samples of *S. invicta/macdonaghi* and *S. richteri* shared 1921 alleles with the SB samples of *S. invicta/macdonaghi*, but only 994 alleles with the SB samples of *S. richteri*, a two-fold difference that is unlikely to be due to incomplete lineage sorting alone (exact binomial test, $p < 10^{-12}$; Supplementary Note 6). We corroborated this result with an additional analysis of the remaining species, using one SB and one Sb sample per species (Supplementary Fig. 12). Finally, we used a phylogenetic weighting method (Twisst[28]; Supplementary Note 7) to quantify the support for alternative topologies in non-overlapping windows of four genes across the supergene. Introgression was the most highly supported topology, found in 2.5 times more windows than the topology in which the supergene is an ancestral trans-species polymorphism (Supplementary Figs. 13 and 14).

Outside the supergene region, the strong support for the species tree is in line with the absence of detectable contemporary gene flow between sympatric populations of *S. invicta/macdonaghi* and *S. richteri* in their native range[29,30]. Particular aspects of fire ant biology ensure at least occasional geographic contact between distinct lineages: flooding transports entire floating fire ant colonies along the major South

American river systems[31,32], while mating flights can reach altitudes of 300 m, with winds carrying young queens over great distances[32]. Importantly, *S. invicta/macdonaghi* and *S. richteri* do hybridize in their invasive North American range[29,30]. This shows that barriers to gene flow between these species are not yet irreversible and makes it plausible that such barriers were weaker in the past. Contact between lineages after allopatric speciation could therefore have facilitated the relatively old introgressions of Sb from *S. invicta/macdonaghi* to other species in our study. However, this explanation is unlikely for the more recent introgression into *S. megergates*, given the current sympatry between this species and *S. invicta/macdonaghi*. Instead, we hypothesize that, for this introgression and perhaps also some older introgression events, a hybrid Sb-carrying queen was accepted into an established multiple-queen colony of *S. invicta/macdonaghi*. This process could have been facilitated by the green-beard behavior of multiple-queen colonies whereby Sb-carrying workers accept Sb-carrying queens. The acceptance of Sb-carrying queens into existing multiple-queen colonies would have provided a crucial social buffer to any allelic incompatibilities of initial hybrid queens (see Supplementary Note 8).

Over long timescales, the retention of introgressed Sb supergene variants suggests that the costs of hybridization are outweighed by the combination of adaptive benefits of Sb in some environments, the green-beard behavior of Sb-carrying workers[13] (in which they only accept Sb-carrying queens), and, potentially, meiotic drive[33]. Future studies of the evolution and function of the social chromosome supergene system will have to take the introgression of Sb into consideration. An important line of research will be to understand the dynamics of selection on *Sb/Sb* queens. Such queens are smaller and less fertile than *SB/Sb* or *SB/SB* queens[34,35]. Reproductive *Sb/Sb* queens are virtually absent in *S. invicta* populations, where the resulting lack of recombination is associated with an accumulation of deleterious mutations[12,14,17,36]. However, *Sb/Sb* queens are intriguingly present in at least some *S. richteri* populations[37], thus enabling recombination between Sb haplotypes. Because of the differences in recombination regime and the bottlenecks associated with introgression, the Sb variants carried by each species may have different mutational loads or have different epistatic interactions with the rest of the genome. The accumulation of deleterious mutations in Sb likely creates a fitness cost that contributes to maintaining the system balanced in *S. invicta*. Indeed, this type of effect has been predicted by theoretical studies of supergene evolution[38,39], and suggested to occur in the supergenes of species like the butterfly *Heliconius numata*[40], the white-throated sparrow[41], the ruff[42] and the seaweed fly[43]. Further studying the dynamics of Sb degeneration will be essential to understand the evolution of the supergene system across all species.

Overall, our results clearly show that Sb evolved in an ancestor of *S. invicta/macdonaghi* that existed after divergence from the lineage leading to *S. richteri*. Through adaptive introgression, the Sb supergene variant spread to *S. richteri*, *S. megergates*, *S. interrupta* and *S. AdRX*. Supergene architecture can thus enable a complex social phenotype encoded by alleles at multiple genes to permeate species boundaries repeatedly. Our finding complements recently documented introgressions of smaller supergenes in other systems[6–9]. The introgression of the Sb supergene variant shows how a species can rapidly gain the genetic basis for a complex phenotype, and how selection for changes in a social organization can overcome the constraints of genomic architecture.

## Methods

**Collection and sequencing of fire ants from across their native range.** We collected and flash-froze *Solenopsis* fire ants from 88 South American colonies

(national collection and export permit from Uruguay N° 001/2015 to M.B, E.S., and Y.W., national and provincial collection and export permit for Argentina [Entre Rios, Santa Fe, Córdoba and Corrientes] N° 007/15, 282/2016, 20.911.387, 025052053-115, 433/02101-0014449-4 and 25253/16 to C.I.P., E.S. and Y.W., national collection and export permit for Brazil N° 14BR015531/DF to M.C.A.). After phenol-chloroform DNA extraction on a pool of ten workers from each colony, we performed a Restriction Fragment Length Polymorphism assay[19] to determine whether both social chromosome variants are present (Supplementary Note 9 and Supplementary Data 1). From each of the 41 colonies that included only the SB variant, we retained one male. From 47 colonies that included both supergene variants, we either retained one SB and one Sb male (for 14 colonies), one Sb male (for 27 colonies), one SB male (for two colonies), two Sb males (for three colonies) or three Sb males (from one colony). We tentatively determined species identity through partial sequencing of the mitochondrial cytochrome oxidase I gene. Two colonies from *S. pusillignis* and *S. saevissima* lacked males; thus, we instead retained a pool of 10 workers. From each of the resulting 107 samples, we prepared an individually barcoded Illumina TruSeq PCR-Free library with 350 bp insert size, which we sequenced on Illumina HiSeq, providing 10.3-fold average genome coverage per sample. We additionally obtained sequence reads from 260 *Solenopsis* males and 1 pool of workers (*S. geminata*) from seven other studies: PRJNA396161[14,17], PRJNA542606[16], PRJNA42136718, PRJNA450756[22], PRJNA182127[14], SRR621118[12], and SRX021921[21]. We performed optical duplicate filtering (minimum optical distance 2500), quality filtering and adapter trimming using clumpify (BBmap, v37.50) and skewer (v0.2.2, --mean-quality 20, --end-quality 15, minimum length of 80 for 100 bp reads and of 100 for 150 bp reads; removing degenerate reads). Supplementary Data 1 provides further details on sample collection and sequencing.

**Genotyping through mapping reads to the reference genome**. We aligned the filtered reads for each sample to the *S. invicta* reference genome assembly Si_gnGA[17] and bait sequences including the *S. invicta* mitochondrion, *phi*X phage and *Wolbachia* sequences (NC_014672.1, NC_001422.1, AF243435.1, AF243436.1, CP001391.1, AM999887.1, GCA_003704235.1) using bwa-mem2 (v2.0pre2 with parameters -B 6 -E 2 -L25,25 -U 50 -T 50 -h 4,200 -a -V -Y -M). Sequence divergence within our dataset is sufficiently low for this approach as, for example, 98.95% of individual reads and 97.16% of pairs of reads from the outgroup species used (*S. geminata*; sequence divergence from *S. invicta* ~3.9 mya[44]), map directly to the *S. invicta* reference genome (Supplementary Data 1). For samples from PCR-based library preparation protocols, we marked duplicate reads with sambamba v0.7.1. We identified and subsequently excluded 23.05 Mb of regions that had coverage higher than median coverage plus 3 standard deviations of single-copy gene coverage across all samples because such high-coverage regions likely represent collapsed repeats (mosdepth v0.2.9, bedtools v2.27.1, bedops v2.4.30). We jointly called variants in all individuals using freebayes (v1.2.0[45], with parameters --min-alternate-count 4 --min-alternate-fraction 0.4 --min-coverage 4 --ploidy 2 --use-best-n-alleles 8 --min-mapping-quality 40 --min-base-quality 30 --use-reference-allele). Although male ants are normally haploid, the genotyping process considered that samples are diploid so that we could identify and remove diploid males and apparently heterozygous sites that could be due to copy-number variation or sequencing artifacts. After decomposing multi-nucleotide variants, we retained only single-nucleotide polymorphisms (SNPs) and excluded low-quality variants (Q < 30), low coverage variants represented by either only forward or only reverse reads, and variants overlapping high-coverage genomic regions or simple sequence repeats (trf v4.09, bcftools v1.10.2). After this first round of genotyping to identify high confidence variants, we performed relaxed targeted genotyping using freebayes --haplotype-basis-alleles of each sample with the following parameters: --ploidy 2 --haplotype-length −1 --use-best-n-alleles 4 --min-mapping-quality 30 --min-base-quality 28 --min-alternate-fraction 0.35 --min-alternate-total 2 --min-coverage 2 --use-reference-allele. After decomposition, individual genotypes where coverage was greater than median plus 3 standard deviations of coverage on single-copy genes or with an allelic balance between 0.25 and 0.75 were set to missing because such sites are not expected in haploids and thus are untrustworthy. For the three pools of workers, we created a haploid consensus by randomly selecting one allele at each heterozygous site. We had initially collected and sequenced 18 additional males (Supplementary Data 1, not included in the sample numbers used above), but for nine of these, more than 25% of variant sites could not be genotyped, and the nine others had particularly high numbers of heterozygous sites indicating that they are likely diploid. All 18 males were removed from further analysis. We retained variant sites for which more than 75% of the remaining 368 samples were genotyped. We analyzed sequences of the *Gp-9* marker (Supplementary Data 1) and performed principal component analysis (Supplementary Fig. 15) to confirm supergene variants.

**Coalescent-based phylogenetic inference**. The *S. invicta* genome assembly (Annotation Release 100) contains exactly one copy of 5851 of the 5991 genes expected to be in a single copy across Hymenoptera (BUSCO v4.0.5[23]). We made an initial consensus tree (which had the same topology as the species tree in Fig. 1e) based on genes mapped to chromosomes 1–15, rooted to *S. geminata*, which we used to label species of each sample according to Yan et al.[18]. Taxonomic

assignment based on a mitochondrial phylogenetic tree was unreliable (Supplementary Note 10 and Supplementary Fig. 16). We retained the most informative 1881 out of 5851 genes for which the 342 males identified as *S. invicta*, *S. macdonaghi* and *S. richteri* (the expected closely related ingroup species in the phylogenetic tree of *Solenopsis*) had a per-gene average of at least ten single-nucleotide differences relative to the reference genome (this step removed non-informative genes that had few or mostly singleton SNPs or that were polymorphic only in the basally diverging species). We obtained a consensus sequence for each of these genes, including coding sequence and introns for each sample using bcftools consensus v1.10.2 from the VCF file of all genotypes. Modeltest-ng v0.1.6[46] identified the best substitution models for intronic sequences and for coding sequences (Supplementary Data 2). For each of the 1881 single-copy genes, we then used RAxML-NG v0.9.0[47] with 50 random and 50 parsimony starting trees to build maximum-likelihood trees with 100 bootstraps, using separate partitions for introns and exons. After removing the trees for the 153 genes that did not converge during the RAxML tree inference, we used ASTRAL-III v5.14.3[24] to create two coalescent-based trees, each supported by 100 bootstraps: one tree based on 97 single-copy genes in the supergene region of social chromosome 16, and one tree based on 1631 single-copy genes from chromosome 1–15. Trees were then rooted to the outgroup *S. geminata*, and extremely short branches (branch length < 0.05) were collapsed into polytomies. The topologies of these two trees were largely congruent with trees constructed from coding sequence alone, or constructed from the full set of 5851 genes with either a coalescent-based tree inference from gene trees (ASTRAL-III) or from concatenation (RAxML-NG, Supplementary Fig. 7 and Supplementary Note 4). To generate Fig. 1e, we used the R package dendextend[48] and subsequently adjusted tree layouts, labels and background colors using Affinity Designer. We dated phylogenetic trees (Fig. 1) using calibrated node ages[44,49] for major lineages in IQ-TREE v2[50] (Supplementary Note 1). The classic ABBA-BABA test[51] is inappropriate for our data because of the topology of the supergene tree. Instead, we performed a similar test, examining the differences in allele sharing between the Sb samples and either the SB samples of *S. invicta/macdonaghi* or the SB of each of the other species (a "BBBAA-BBABA" test; Supplementary Note 6).

**Reporting summary**. Further information on research design is available in the Nature Research Reporting Summary linked to this article.

## Data availability

All DNA sequences generated in this study have been deposited in the Sequence Read Archive database under accession code PRJNA685290 (individual accession numbers are detailed in Supplementary Data 1). The VCF genotype matrix and the de novo assemblies we generated are available at https://wurmlab.com/data/supergene_introgression. The source data used to produce each figure, including Newick tree files, are available on GitHub at https://github.com/wurmlab/2021-fire-ant-social-supergene-introgression, with additional information supplied as Supplementary Data.

## Code availability

Code and analysis scripts are available on GitHub at https://github.com/wurmlab/2021-fire-ant-social-supergene-introgression, and at https://wurmlab.com/data/supergene_introgression.

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

## Acknowledgements

This work was supported by the European Union (FP7 Marie-Curie-Fellowship PIEF-GA-2013-623713 to E.S. and Y.W.), the German Academic Exchange (DAAD 570704 83 to E.S.), the Fundação de Amparo à Pesquisa do Estado de São Paulo (FAPESP BEPE grant 2014/04943-0 to M.C.A.), COLFUTURO (to G.L.H.), the Conselho Nacional de Desenvolvimento Científico e Tecnológico (Ciência sem Fronteiras grant 248391/2013-5 to Y.W.), the Biotechnology and Biological Sciences Research Council (grants BB/K004204/1, BB/T015683/1, and BB/S507556/1 to Y.W.) and the Natural Environment Research Council (grant NE/L00626X/1 to Y.W.). We thank S. Coelho, D. Pereira Nogueira Da Silva, N. Curi, N. Souza Araujo (Universidade de São Paulo), R. Jaffé (Instituto Tecnológico Vale), Pablo Álvarez Osorio (Buenos Aires), E. Boné (Universidad de Buenos Aires), Y. Guillij (Dir. de Usos Sust. de los Recursos Naturales, Entre Ríos), Dir. de Conservación de la Biodiversidad Argentina, C. Durrant, M. dos Reis, L. Rodrigues Santiago, C. Martínez-Ruiz, R. Nichols, and C. Eizaguirre (Queen Mary University of London), for their help with organization, sampling, permits, preparation, sequencing or analysis, useful discussions, and comments on the manuscript.

## Author contributions

Initial conceptualization: E.S., Y.W. Experimental design: R.P., E.S., F.L.-O., Y.W. Sample collection, identification, and processing: E.S., C.C.-C., M.C.A., C.I.P., M.B., Y.W. DNA extraction and sequence library construction: E.S., C.C.-C.. Initial data analysis: E.S., R.P., A.P. Main data analyses: E.S., R.P., Y.W. Additional data analyses and visualization: E.S., R.P., M.K.P., G.L.H., A.P., F.L.-O. Supervision: Y.W. Writing—original draft: E.S., R.P., Y.W. Writing—review and editing: R.P., E.S., M.K.P., G.L.H., A.P., F.L.-O., Y.W.

## Competing interests

The authors declare no competing interests.
