## [Peer Review File · Nature Communications]

Recurring adaptive introgression of a supergene variant that determines social organizationReviewers' Comments:

Reviewer #1:

Remarks to the Author:

The authors aim to add evidence in support or opposition to the fundamental importance of two key concepts in modern evolutionary biology: introgressive hybridization and supergenes.

As this is a short article, there is not overwhelming evidence in support of their claims, but one can clearly follow the logic and understand their basis. I do have a few concerns, which may be solely due to space constraints but I think should be addressed.

First, on the evidence that the Sb haplotype was introgressed multiple times. The phylogeny was generated according to best practices, and is well presented. The authors clearly lay out the competing hypotheses in their figure. However, a critical node in their Sb phylogeny has slightly less support than the rest. It is often difficult to have complete confidence in the phylogeny of an inversion, because regardless of how many loci are contained within it, there is likely only a single common ancestor of the inversion itself. Therefore, it is quite difficult to be sure that the inversion actually follows scenario D as opposed to B (incomplete lineage sorting). I do, however believe based on the phylogeny that there has been at least some introgression, including the megergates individual that has a haplotype extremely similar to its potential introgression partners. In fact, branch lengths would be a good way to provide additional evidence. Edelman et al (2019) provide a statistic (QuIBL) that considers the internal branch lengths of potentially introgressive phylogenies to explicitly distinguish between ILS and introgression.

As far as the supergene, the authors provide only circumstantial evidence that this large inversion is in fact a supergene. Perhaps elsewhere in the literature there is evidence that multiple genes within this inversion are important to the multi-queen phenotype discussed here, but one should not assume that large inversions necessarily capture several related genes. Inversions can also capture cis-regulatory transcription modifiers, and break genes in the middle (rendering them inactive) as means of impacting phenotypes. However, this is a somewhat minor point, given the main thrust of the paper is about introgression rather than the genetic structure of the supergene itself.

Reviewer #2:

Remarks to the Author:

This manuscript by Stolle et al. lays out the argument that a supergene variant spread through several closely related species via adaptive introgression. This argument contradicts the assertion of several other recent studies which posited that the two alternative supergene variants were present in the common ancestor of at least *S. invicta* and *S. richteri*. In general, I think that the authors make a compelling case for the role of introgression in spreading the Sb haplotype amongst closely related species, but I have some misgivings about several of their other major inferences, as well as some minor suggestions.

The coalescent-based phylogenetic approach shows pretty clearly that differentiation between species-specific SB haplotypes predates differentiation between species-specific Sb haplotypes, providing good evidence for a role of introgression in spreading the Sb variant across species boundaries. However, the authors go on to argue that this provides evidence that the Sb variant introgressed from *S. invicta* into *S. richteri* and several other species at least five times. This assertion is built on a verbal argument, and I don't think that their analysis allows them to determine the direction of introgression for the following reasons:

1. The coalescent-based phylogenetic approach relies on the assumption the the Sb haplotype does not recombine. In fact, the authors mention the presence of recombination in Sp/Sp individuals in *S.*

richteri. Moreover, occasional recombination or gene conversion from SB to Sb in any of the species is possible. On the phylogenetic time-scale, such events could mask the evolutionary history of the Sb haplotype in the context of a coalescent analysis.

2. The SRR9008168_xAdR-135-Pos-littleb individual actually appears to be sister to all other Sb individuals in figure 1e. They only had one Sb individual of this species, but this position in the phylogeny at least raises the question of Sb evolving first in this xAdR species and spreading into *S. invicta*. At the very least, the authors' verbal model explaining their rationale for predictions about the direction of introgression needs to be expanded and clarified. I understand why the ABBA BABA test is not the best fit for the dataset here, but other formal tests of introgression would greatly improve support (or not) for their proposed scenario. If they can't add support, I suggest that they acknowledge that the direction of introgression is unknown.

3. I wonder whether there is any information about which groups of Sb variants can still recombine (in which groups do Sb/Sb individuals reproduce)? I would find it slightly surprising if the Sb is less degenerated in the two *S. richteri* clades compared to the adjacent *S. invicta* clades, given the implied direction of introgression.

In summary, there is an important and credible finding here: introgression of a supergene haplotype is likely in this system, calling into question recent conclusions about the evolutionary history of this supergene system. However, the authors don't provide convincing evidence about the direction of introgression; this would require additional analysis and may be limited by the small sample size of the outgroup species that have the SB and Sb genetic polymorphism.

Additional comments:

I found the initial question posed in the introduction ("Is it possible for a complex trait involving alleles at many loci to evolve in one species and then introgress into another?" to be overly naive in light of our understanding of other sex chromosome/supergene systems. In fact we know the answer to this question based on work from butterfly supergenes and sex chromosomes, for example (some relevant studies are cited at the very end of the discussion, but I think they should be properly acknowledged at the outset, rather than treating this like a completely novel question). ** I would add a sex chromosome example, e.g. 10.1093/molbev/msy181 (there are other examples).

Lines 140-142: The authors suggest that adaptive benefits of Sb outweigh the costs of hybridization. In my opinion, this possibility is complicated by the selfish nature of Sb, with both green beard and meiotic drive contributing to its persistence in *S. invicta*. The sentence implies that it's adaptive at the organismal level, but I don't think we know this.

Out of curiosity, what is going on with the 153 genes that did not converge in the coalescent-based phylogenies? Are there additional signatures of past hybridization genome wide?

Line 242: ABBA-BABBA should be ABBA-BABA

It would be interesting to see the source of the samples in the trees, since sampling localities are widely distributed and include native and invasive ranges. Could this be added in a supplementary figure?

Reviewer #3:

Remarks to the Author:

In this manuscript, the authors use a large-scale genomic dataset to infer the evolutionary history of the social supergene of the *Solenopsis* ants. By comparing a genome-wide phylogeny with a

phylogeny made only with genes from the supergene region, they convincingly show that the supergene variant Sb was introgressed multiple times between the species of the clade.

The biological system studied here is very attractive, and the results described by the authors in this simple study are interesting as they support the fact that inversions can act as “adaptive cassette”, allowing combinations of coadapted alleles to be introgressed as a single unit, and therefore to be protected from recombination. Moreover, this study raises interesting questions on the evolution of supergenes and inversion polymorphisms. The analyses seem to have been done correctly and the literature is properly referenced (but additional references to other supergenes with introgressed alleles are necessary. e.g. white-throated sparrow supergene, *Papilio* supergenes, etc.)

Overall, I found the manuscript very well written and I appreciated the effort made by the authors to provide a simple manuscript supporting an interesting conclusion. Nevertheless, I found this study lacks details and analyses to properly support the conclusions and assess their validity. This is especially true as this study contradicts recent finding from another team (Yan et al., 2020). Hence, I think that performing additional analyses (detailed below) would strongly benefit to this work by providing more support to its conclusion. If these additional analyses are concordant with the analyses provided in this manuscript (which I do not doubt), I think this work will be significant to the field and be of general interest.

-Major and Minor comments-

Major:

1) In this manuscript, the authors consider the supergene region as a single genomic unit. However, previous studies have shown that it includes three chromosomal inversions. Moreover, Yan et al., 2020, have shown that there are likely recombinations between SB and Sb haplotypes. This means that different regions of the supergene might have different evolutionary histories. Therefore, instead of providing a single phylogeny for the whole supergene, I would find very interesting and informative to observe how the phylogenies vary along the genome and especially along the supergene. For instance, by computing the support for the different topologies along the genome (e.g. using Twisst), as done in Jay et al., *Current Biol.*, 2018, Edelman et al., *Science*, 2019 or Fontaine et al., *Science*, 2015, for example. This could help to understand the discrepancy between Fig 1 and Fig S4, and with the analyses by Yan et al. 2020.

2) The authors seem to consider that Sb evolved first in *S. invicta* and was then introgressed in other species. I do not see from which analysis this conclusion is derived. I guess this is because Fig. 1 show that the Sb males from other species are nested within the *S. invicta* Sb males. This should be mentioned in the manuscript. Moreover, I think this is not that clear. For instance, Fig 1 seem to show that the *S. AdRX* Sb male is paraphyletic with other males from the Sb clade. This could mean that the Sb allele could have arisen in *S. AdRX* and then introgressed to *S. invicta*. In addition, since Sb and SB do recombine, the position of the samples within the phylogeny could reflect the effect of these recombinations and not only of the introgression events (as suggested by Yan et al. 2020). For instance, the Sb haplotype could have arisen in *S. richteri* and then introgressed to *S. invicta*. Recombination between Sb and SB in *S. invicta* could create this pattern of *S. richteri* males nested in *S. invicta*. Additional analyses are required to support the direction of introgression and this should in particular involve sliding window phylogeny or similar analyses (as I suggested before)

3) The topology of the *S. richteri* clade in the whole-genome phylogeny (Fig 1) is very odd. In particular, the position of the samples involved in the introgression event 1 is not normal (for instance, the branch length is far from being poisson-distributed). This peculiar pattern is not visible in fig S4. I wonder about the origin of this particular pattern and I think that the analyses proposed previously might shed some light on it.

4) It would be very interesting to perform a datation of the phylogenies, which could allow to infer the time of the introgression events (as before, it should not be performed on the whole supergene since

different segments of the supergene might have different histories)

5) From my point of view, the title of the paper is problematic. A supergene is a locus. A locus cannot be introgressed, only an allele can. Therefore, I think a correct title should be : "Recurring adaptive introgression of a supergene ALLELE that determines social organization" or "Recurring adaptive introgression AT a supergene that determines social organization". This issue is present in several part of the manuscript. For instance, in "These supergenes can spread among populations unhindered by recombination, but the extent to which they introgress across species boundaries remains unresolved." or in "The social supergene evolved in one species and subsequently introgressed into others". The authors do not talk about the introgression of inversions but this is exactly what it is (in the two previous sentences, changing "supergene" to "inversion" would be perfectly correct). The author said that only the Sb haplotypes have been introgressed and not the SB haplotypes, so the whole supergene has not been introgressed. Literally, the supergene have not evolved a single time. The inversions associated with the Sb variant have evolved a single time and have been recurrently involved in supergene formation following their introgression (because they were maintained polymorphic). For me, this study show the recurrent introgression of three inversions associated with a peculiar social organization, which led to the recurrent formation of a supergene underlying polymorphic social organization. This resemble to the situation modeled by Jay et al. BioRxiv 2020. I think it is very important to be very clear on what is a supergene, and for me, a supergene is a set a tightly linked loci segregating together due to recombination suppression. Therefore, I do not think that an inversion is a supergene, it is only an allele of a supergene. I believe talking about the recurrent formation of a supergene following introgression of same inversions would make this study of even more general interest. It raises questions about the force maintaining these inversions polymorphic, which is very interesting considering the ubiquity of inversions in genomes and their recurrent association with complex adaptation.

Minor:

6) Many species are considered as "outgroups", which is non-sense since some of them are involved in introgression events. I think only the most divergent species (*S. geminata*) should be considered as "outgroup", it would make the manuscript more clear.

7) I think the Fig 1e is hard to read. The colors of *S. invicta* and the outgroups are too similar. The sample labels are useless since we cannot read them. I am not sure it is very useful to have background colors for the part of the figure showing the links between the two phylogenies, since it hampers to properly see the links. In addition, for some samples, the link goes all the way to the sample labels (Outgroup and *S. richteri*), whereas for others it is limited to the middle part of the figure (*S. invicta*). This makes the figure look a bit strange. I'm not sure it will improve the figure, but can you try with the same link length for all samples ?

8) "Once hybridization occurred, we hypothesize that the type of "green-beard" phenotype encoded by Sb facilitated the acceptance of hybrid Sb-carrying queens into established multiple-queen colonies of *S. invicta/macdonaghi*." This sentence is hard to follow and I suggest it could be rephrased. I do not think that referring to the "green-beard" hypothesis is necessary here, as this theory is still controversial and referring to it adds unnecessary complexity

9) "Our finding complements recently documented introgressions of smaller supergenes encoding simpler phenotypes (12, 29–31) in other systems." Beside the terminology issue raised in (5), I think the use of "simpler" is abusive here, as it implies arbitrary considerations of complexity. For instance, Dixon et al., MBE, 2019 have shown that an introgression has led to the formation of a neo sex chromosome, and sex clearly cannot be considered as a "simpler" phenotype.

10) "included an average of at least 10 single-nucleotide differences relative to the reference genome."

Not very clear for me what "an average of at least" means. Can you rephrase this sentence ?

11) "we used ASTRAL-III v5.14.3 (19) to create two coalescent-based trees supported 100 bootstraps". Not very clear. You created two coalescent-based trees with 100 bootstraps ?

12)L31: "it is possible that its emergence (12) or spread (8, 11, 13)". As I mentioned in my comment (5), for me, emergence and spread are here exactly the same thing: the introgression of inversions which were subsequently maintained polymorphic.

13) "be robust to tree discordance due to processes including tree reconstruction errors, incomplete lineage sorting and gene conversion". Tree discordance due incomplete lineage sorting and gene conversion (and recombination) are informative since they reflect the true history of the sequences studied. I do not think that providing a single tree erasing these differences makes sense

14) L128-137: You discuss here why the Sb variant was successful in establishing in other species. I think it could also be very interesting to quickly discuss here why it remains polymorphic, i.e. why the Sb variant did not fix in populations.

REVIEWER COMMENTS

Reviewer #1 (Remarks to the Author):

The authors aim to add evidence in support or opposition to the fundamental importance of two key concepts in modern evolutionary biology: introgressive hybridization and supergenes.

As this is a short article, there is not overwhelming evidence in support of their claims, but one can clearly follow the logic and understand their basis. I do have a few concerns, which may be solely due to space constraints but I think should be addressed.

>>> Authors 1.0: We thank the reviewer for their constructive comments and suggestions for improvement. Below you find a point-by-point response. We hope the reviewer agrees that the new analyses that we added at the recommendation of all three reviewers considerably strengthen the evidence for the introgression of Sb between species. We appreciate that the main text condenses a lot of information into a small article. This is in line with the article format. We feel that it also is appropriate for us to highlight that the Supplementary Materials section is now 32 pages long. The Supplementary Materials provide extensive additional evidence that could not fit into the main text, including additional analysis details, results, figures, and discussion.

Reviewer 1.1: First, on the evidence that the *Sb* haplotype was introgressed multiple times. The phylogeny was generated according to best practices, and is well presented. The authors clearly lay out the competing hypotheses in their figure. However, a critical node in their *Sb* phylogeny has slightly less support than the rest. It is often difficult to have complete confidence in the phylogeny of an inversion, because regardless of how many loci are contained within it, there is likely only a single common ancestor of the inversion itself. Therefore, it is quite difficult to be sure that the inversion actually follows scenario D as opposed to B (incomplete lineage sorting). I do, however believe based on the phylogeny that there has been at least some introgression, including the megergates individual that has a haplotype extremely similar to its potential introgression partners. In fact, branch lengths would be a good way to provide additional evidence. Edelman et al (2019) provide a statistic (QuIBL) that considers the internal branch lengths of potentially introgressive phylogenies to explicitly distinguish between ILS and introgression.

>>> **Authors 1.1:** The reviewer raises an important point which could potentially affect conclusions regarding introgressions to *S. richteri* (but not the introgression events into the three other species): in the ASTRAL tree (Fig. 1e), the node that separates the clade containing the *S. richteri* SB samples from the clade containing *S. invicta/macdonaghi* SB samples and all *Sb* samples has 98% bootstrap support and 0.89 posterior probability, which is less than some of the other key nodes in our phylogeny. This node is indeed important because for subset of samples from *S. richteri* and *S. invicta/macdonaghi*, it distinguishes the ancestral polymorphism scenario (the tree on the left of the figure below) and the introgression scenario (the tree on the right of the figure below; these figures are taken from Supplementary Fig S12a).

Our overall conclusion that the *Sb* supergene variant introgressed into the three early diverging species is independent of the patterns at this node. We also performed additional analyses that are complementary to our original phylogenetic analysis (outlined further down). Nevertheless, **three lines of evidence directly based on our phylogenies support the topology as shown.** First, the ASTRAL-III posterior probabilities for the two alternate quartet topologies at this node are very low (0.1 and 0.01). Second, a statistical test for polytomy (ASTRAL-III) indicated that there is no evidence for a polytomy at this node ($p < 0.05$). Furthermore, the independent RAxML tree we produced from concatenated alignments shows unambiguous support for the node in question (100%, Supplementary Figure S7). Overall, we can thus conclude that the topology at this node is well supported, *i.e.*, our tree is accurate, but the underlying data may be noisy.

There are two possible causes for this noise. The first is that the internal branches separating *S. AdRX*, *S. richteri* and *S. invicta/macdonaghi* are relatively short, as shown by the trees we made for chromosomes 1 to 15 for our population data by ASTRAL (Fig. 1e) and independently using RAxML (concatenated data, Suppl. Fig. S7). Internal short branches are expected to lead to incomplete lineage sorting (ILS), which reduces the phylogenetic signal in the tree. The effect of ILS would be stronger for the supergene than the rest of the genome, because the supergene contains far fewer genes and fewer samples (in the rest of the genome, the separation between two species takes all samples into account; in the supergene region, only the SB samples are informative). As a result, we would expect ILS to have a particularly strong effect on the branch separating the speciation event from the node representing the supergene origin (illustrated in the figure above). A second plausible source of noise is historical introgression of alleles between the SB variants of *S. invicta/macdonaghi* and *S. richteri*. Indeed, the discordance between the species tree and the mitochondrial genome phylogeny (Suppl. Text S9) suggests that some historical introgression has occurred (we did not investigate this further as it is outside the scope of this study). In summary, we believe our tree to be correct, but that ILS and, possibly, past introgression have reduced the signal supporting the separation of the SB samples between species.

In light of the questions of this and other reviewers, we have performed **four additional analyses to test our inference of introgression**, all of which pertain to the node highlighted here. First, we produced a dated species tree (chromosomes 1 to 15) from >2100 genes taken from *de novo* assemblies instead of SNP calls, to ensure that our species phylogeny is correct (Supplementary Figure S4). As expected, we found the same species topology as with ASTRAL (Fig. 1e) and with RAxML (Supplementary Figure S7). Incidentally, the node separating *S. invicta/macdonaghi* and *S. richteri* had slightly less than 100% bootstrap support, supporting our inference that the signal separating these two species is noisy even outside the supergene.

Second, we followed Reviewer 1's suggestion and used the QuIBL statistic to explicitly distinguish between introgression and ancestral polymorphism and ILS (Edelman et al. 2019) for the supergene region. We performed two separate analyses, one testing the introgression between *S. invicta/macdonaghi* and *S. richteri* (Supplementary Text S5, Supplementary Figures S8-9), the other testing the introgression between *S. invicta/macdonaghi* and the earlier diverging clades (Supplementary Text S5, Supplementary Figures S10-11). In both analyses, QuIBL supports the introgression scenario over the scenario of ancestral polymorphism and ILS.

Third, we used the method Twisst (suggested by Reviewer 3.1), which quantifies the support for every possible topology over sliding windows across the supergene region (Supplementary Text S7, Supplementary Figures S13-14). This analysis showed that the topology representing introgression (*S. invicta* to *S. richteri*) is the most commonly supported topology in the supergene (found in 51% of windows), followed by the topology representing ancestral polymorphism topology (26% of windows) and the topology representing introgression with the opposite direction (*S. richteri* to *S. invicta*, 13% of windows). The strong relative support we find of the introgression topology compares favorably with what is typically seen in other applications of Twisst (Jay et al. 2018, Current Biology; Moest et al. 2020, PloS Biology; Dixon et al 2019, Mol Biol Evol).

Fourth, we tested whether the numbers of alleles shared by Sb and the SB of each species was the same, as expected under the scenario of ancestral polymorphism and ILS (effectively a BBAA-BBABA test, as explained in Supplementary Text S6). This was not the case.

Instead, two times more alleles were shared between Sb and the SB of *S. invicta/macdonaghi* than between Sb and the SB of *S. richteri*, as expected under the introgression scenario. We found analogous patterns in comparisons between *S. invicta/macdonaghi* and each of the other species.

These results are summarized in the main text of our manuscript (from line 147). We hope that the reviewer agrees that, with the additional analyses we performed, our data overwhelmingly supports the introgression scenario. We thank the reviewer for asking us for extra details on this point, as we think our new analyses make our study much stronger.

Reviewer 1.2: *As far as the supergene, the authors provide only circumstantial evidence that this large inversion is in fact a supergene. Perhaps elsewhere in the literature there is evidence that multiple genes within this inversion are important to the multi-queen phenotype discussed here, but one should not assume that large inversions necessarily capture several related genes. Inversions can also capture cis-regulatory transcription modifiers, and break genes in the middle (rendering them inactive) as means of impacting phenotypes. However, this is a somewhat minor point, given the main thrust of the paper is about introgression rather than the genetic structure of the supergene itself.*

>>> Authors 1.2: Multiple definitions of “supergene” exist. A widely used definition refers to multiple linked functional genetic elements that allow switching between discrete phenotypes that are maintained in a stable local polymorphism (Thompson and Jiggins 2014 *Heredity*). The fire ant social chromosome leads to two discrete phenotypes under local polymorphism. Functional genomic work in the fire ant has so far been unable to unambiguously demonstrate how particular genes contribute to the multiple phenotypic differences between social forms or between Sb/SB and SB/SB individuals. We agree that selection affecting two genes or even an inversion disrupting just one gene can lead to supergene-like inheritance. However, substantial evidence suggests that several genes contribute to the phenotypic differences. For example, the coding sequences of some genes involved in pheromone signaling or processing are intact in one supergene variant and degraded or lost in the other (e.g., Wang et al. 2013 *Nature*, Pracana et al. 2017 *Evolution Letters*, Cohan et al. 2018 *GBE*). Furthermore, some differences in gene expression between the two supergene variants emerged in response to antagonistic selection (Martinez-Ruiz et al. 2020 *eLife*, Arsenault et al. 2020 *Mol Ecol*). These gene expression differences imply the presence of multiple co-adapted elements (even if these were only selected *after* the original emergence of the inversion). Finally, even if disruption of a single gene were responsible for the social phenotype, the reduced fertility of queens and males carrying the Sb supergene variant are likely to be due to degradation of other genes in the supergene region. We agree that this manuscript is not the appropriate venue to discuss supergene definitions at length. However, to help reduce ambiguity, we now provide additional details regarding the molecular bases of the supergene-associated phenotypes in the Introduction (line 28):

“The supergene region contains more than 470 protein-coding genes, most of which are present in both supergene variants¹². The duplication or absence of some genes in Sb^{14,15}, and the divergence in sequence and expression of several others suggest that they contribute to the social phenotype¹⁶. Additionally, Sb differs from SB by three large inversions, and Sb has expanded through the accumulation of repetitive elements^{17,18} (estimated sizes are 20.9 Mb for SB and 27.5 Mb for Sb¹⁷).”

Reviewer #2 (Remarks to the Author):

*This manuscript by Stolle et al. lays out the argument that a supergene variant spread through several closely related species via adaptive introgression. This argument contradicts the assertion of several other recent studies which posited that the two alternative supergene variants were present in the common ancestor of at least *S. invicta* and *S. richteri*. In general, I think that the authors make a compelling case for the role of introgression in spreading the Sb haplotype amongst closely related species, but I have some misgivings about several of their other major inferences, as well as some minor suggestions.*

>>> Authors: We thank the reviewer both for their supportive comments as well as their constructive feedback.

*The coalescent-based phylogenetic approach shows pretty clearly that differentiation between species-specific SB haplotypes predates differentiation between species-specific Sb haplotypes, providing good evidence for a role of introgression in spreading the Sb variant across species boundaries. However, the authors go on to argue that this provides evidence that the Sb variant introgressed from *S. invicta* into *S. richteri* and several other species at least five times. This assertion is built on a verbal argument, and I don't think that their analysis allows them to determine the direction of introgression for the following reasons:*

Reviewer 2.1: *The coalescent-based phylogenetic approach relies on the assumption the the Sb haplotype does not recombine. In fact, the authors mention the presence of recombination in Sp/Sp individuals in *S. richteri*. Moreover, occasional recombination or gene conversion from SB to Sb in any of the species is possible. On the phylogenetic time-scale, such events could mask the evolutionary history of the Sb haplotype in the context of a coalescent analysis.*

>>> Authors 2.1: The reviewer makes a relevant point that recombination and gene conversion can mask true phylogenetic signals. However, our inference is robust because potential recombination would not create the observed pattern, but would only reduce our ability to detect it.

There are two important types of recombination to consider. The first is the Sb-Sb recombination that the reviewer first refers to specifically. As we say in the main text (reworded for increased clarity), Sb-Sb recombination is rare, given that reproductive *Sb/Sb* queens have only been documented in *S. richteri* (line 192):

“Reproductive *Sb/Sb* queens are virtually absent in *S. invicta* populations, where the resulting lack of recombination is associated with an accumulation of deleterious mutations. Intriguingly, *Sb/Sb* queens are present in at least some *S. richteri* populations [...]”

Briefly, the absence of reproductive *Sb/Sb* queens precludes recombination in Sb in extant *S. invicta/macdonaghi*, although such recombination may have happened historically (this is discussed more at length in **Authors 2.3**). The key point, however, is that the occurrence of **Sb-Sb recombination in any species would not change how we interpret our results**. As long as the recombination occurs only among individuals of the same species, we would still expect Sb to cluster by species, regardless of which of the scenarios is correct.

The second type of recombination to consider is between SB and Sb. This occurs at low rates (Ross and Shoemaker, 2018 BMC Genomic Data). Although this type of recombination could affect the phylogenetic signal, it would do so in the opposite direction to what we observe. Recombination would make the Sb variant of each species gradually more similar to its

species-specific SB, *i.e.*, *S. richteri* Sb would cluster with *S. richteri* SB, *S. invicta* Sb would cluster with *S. invicta* SB and other species Sb would cluster with their respective SB. The resulting tree topology would be consistent with a parallel origin of the supergene (Fig.1c), even if polymorphism was ancestral to speciation. The tree topology would not resemble the “introgression” supergene tree topology for which we find overwhelming support in our study.

The point raised here is also in line with the point made in Reviewer 3.1, that our ASTRAL analysis aggregates information from all genes in the supergene into a single phylogeny, potentially masking region-specific processes (such as SB-Sb recombination, particularly if the repression of recombination is not equally effective across the whole region). In response, we have performed a new analysis (topology weighting across the supergene using the Twisst software). The main text now summarizes the results of this analysis (line 162):

“Finally, we used a phylogenetic weighting method (Twisst²⁸; Supplementary Text S7) to quantify the support for alternative topologies in non-overlapping windows of four genes across the supergene. Introgression was the most highly supported topology, found in 2.5 times more windows than the topology in which the supergene is an ancestral trans-species polymorphism (Supplementary Fig. S13-14).”

The most often supported topology was the introgression from *S. invicta/macdonaghi* to *S. richteri* topology (found in 51% of windows). The topology which would be expected by SB-Sb recombination (the “Species tree” topology in Supplementary Fig. S14) was less supported than any other of the scenarios that we considered (present in only 8% of windows; see Supplementary Text S7). A tantalizing possibility is that the phylogenetic signal in these windows reflect SB-Sb recombination; we do not discuss this possibility in our manuscript as it is quite speculative.

A final point is that historical *Sb-Sb* recombination could have masked the signal of historical SB-Sb recombination by mixing SB alleles into the Sb background (and *vice versa*). Such recombination might explain why we still have well supported (if rather shallow) phylogenies despite possible SB-Sb recombination.

Reviewer 2.2: *The SRR9008168_xAdR-135-Pos-littleb individual actually appears to be sister to all other Sb individuals in figure 1e. They only had one Sb individual of this species, but this position in the phylogeny at least raises the question of **Sb evolving first in this xAdR species and spreading into S. invicta**. At the very least, the authors’ verbal model explaining their rationale for predictions about the direction of introgression needs to be expanded and clarified. I understand why the ABBA BABA test is not the best fit for the dataset here, but other formal tests of introgression would greatly improve support (or not) for their proposed scenario. If they can’t add support, I suggest that they acknowledge that the direction of introgression is unknown.*

>>> Authors 2.2: The main evidence that Sb evolved from an ancestral chromosome of *S. invicta* is the position of Sb as a sister clade to SB in *S. invicta* after the separation between this species and all other fire ants (regardless of the internal position of AdRX within the Sb clade). This topology is seen in both the consensus tree we built with ASTRAL (Fig. 1e) and the tree we built from a concatenated alignment (Supplementary Text S4, Supplementary Fig. S7). We realize that we could have made this argument clearer in our manuscript (see Authors 3.2), so we have now rewritten a key part of the relevant paragraph (line 72):

“Instead, the grouping of Sb males as sister to the *S. invicta/macdonaghi* SB clade shows that the Sb supergene variant originated in chromosome 16 of *S. invicta/macdonaghi* after the divergence between this species and *S. richteri*. The presence of Sb sequences from other species within the *S. invicta/macdonaghi* Sb clade shows that Sb introgressed from *S. invicta/macdonaghi* into the other species (scenario in Fig. 1d). ”

Furthermore, we have now added four analyses that support the origin of Sb in *S. invicta/macdonaghi* and not in any other species (lines 147–166) in the main text, as summarized in Authors 1.1). Two of these analyses are particularly relevant for the specific question asked by Reviewer 2. The first is the topology weighting analysis with Twisst. This method explicitly compares the level of support for alternative topologies in windows across the genome. The most common topology across the supergene supported the introgression, from *S. invicta/macdonaghi* to *S. richteri* (51% of windows), and a much smaller number supported the introgression from *S. richteri* to *S. invicta/macdonaghi* (13% of windows) (Supplementary Figure S14, Supplementary Text S7).

For the second analysis, we asked which species had an SB variant that shared most alleles with Sb (Supplementary Text S6, summarized in Authors 1.1). This species can be inferred to be the origin of the supergene. We found that the SB variant of *S. invicta/macdonaghi* shares substantially more alleles with Sb than any of the other species, including *S. AdRX*.

Details for the two other analyses are discussed in the main text (lines 154–166) and explained in the response to Authors 1.1.

In summary, we have overwhelming evidence that Sb is a sister clade of SB in *S. invicta* (as explained here and in Authors 1.1). The most parsimonious conclusion from this is that Sb originated from multiple inversions of the *S. invicta* chromosome 16 (the inverted variant becoming Sb, the non-inverted variant becoming SB) and not in any other species. The placement of the Sb *S. AdRX* sample at the base of the Sb clade would therefore imply that Sb introgressed into this species soon after its origin, rather than originating in this species. Theoretically, the origin of the Sb and the introgression could have occurred concurrently, if the region became inverted in a hybrid between *S. invicta* and *S. AdRX*. This is unparsimonious, as it would require a secondary introgression of Sb into *S. invicta*.

Perhaps more importantly, evidence that the *AdRX* sample is actually a sister species to all the Sb samples is not particularly strong (indeed, we show it as a paraphyletic branch in the tree). In the tree produced from concatenated alignments (Supplementary Figure S4) this *AdRX* sample is placed as the sister to a monophyletic group of individuals (AR186-1, AR186-3, AR187-1, AR187-5, SRR9008173_mac-146, SRR9008257_mac-147), which are also a monophyletic group in the main tree. This group is relatively basal in both trees. Therefore, although we have no evidence that the Sb variant of *AdRX* is a sister clade to all other Sb samples, we expect that this variant has introgressed into *AdRX* quite early in the evolution of the system. We now discuss this sample in the discussion (line 144):

“Additionally, the Sb sample from *S. AdRX* (group 6 in the coalescent-based tree, Fig. 1e) is placed as a sister taxon to a relatively basal clade of Sb in the concatenation-based tree, suggesting that the introgression into *S. AdRX* occurred relatively early”

As an aside, we considered using the DFOIL method (Pease and Hahn, 2015 Systematic Biology), an expansion of the ABBA-BABA D-statistics for symmetrical five-taxon trees (the ABBA-BABA test works only for 4-taxon trees). In principle, this method should allow us to test for direction of introgression between Sb on a species tree such as this one:

((AdRX_SB, AdRX_Sb), (invicta_Sb, invicta_SB)), outgroup)

However, DFOIL (and the ABBA-BABA test) is designed to test the introgression of alleles from one taxon into the genome of another taxon (with recombination). In our study, we are instead considering the introgression of a whole chromosomal region between taxa, which gives us a tree that is extremely discordant from the tree above. As we expected, our attempt at using this software did not work, giving us a combination of D-statistics that does not follow any of the expectations laid out by Pease and Hahn (2015). We left out this analysis from the manuscript as it was inadequate given the nature of our data.

Overall, we hope the reviewer agrees that the analyses that we now present show overwhelming support for the origin of Sb from the ancestral chromosome of *S. invicta* and not of any other species. We also hope that the clarifications that we added to the main text make it explicit that the relatively basal position of the *S. AdRX* Sb sample in the Sb clade is evidence of an early introgression, but not that Sb originated in that species.

Reviewer 2.3: *! wonder whether there is any information about which groups of Sb variants can still recombine (in which groups do Sb/Sb individuals reproduce)? ! would find it slightly surprising if the Sb is less degenerated in the two S. richteri clades compared to the adjacent S. invicta clades, given the implied direction of introgression.*

>>> Authors 2.3: We now summarize the current knowledge of the system in the main manuscript (line 192), having added more details (see Authors 2.1):

“Reproductive Sb/Sb queens are virtually absent in *S. invicta/macdonaghi* populations, where the resulting lack of recombination is associated with an accumulation of deleterious mutations. However, Sb/Sb queens are intriguingly present in at least some *S. richteri* populations³⁶, thus enabling recombination between Sb haplotypes.”

The cause for the absence of *Sb/Sb* queens is unknown. The likely reason is that Sb has remained at low frequency in the population for long enough (with *Sb/Sb* queens being very rare), leading to an accumulation of deleterious mutations that eventually led the variant to become homozygous lethal. We were surprised that the species where Sb recombination occurs (*S. richteri*) is the one where Sb introgressed into, whereas the original species (*S. invicta/macdonaghi*) has lost Sb recombination. This pattern would suggest that the homozygous lethality of Sb in *S. invicta/macdonaghi* evolved after the introgression into *S. richteri*.

Related to this specific question by Reviewer 2, Reviewer 3 asked us to produce a dated phylogeny to have some insight into the timing of the evolution of the system (see Authors 3.4). We have added a new dating analysis, which places the introgression of Sb into *S. richteri* soon after its origin in *S. invicta/macdonaghi* (line 95):

“A divergence time inference (Supplementary Text S3) suggests that the separation of SB and Sb occurred 0.97 million years ago (Mya), after the split between *S. richteri* and *S. invicta/macdonaghi* (1.01 Mya) and before the diversification of *S. invicta/macdonaghi* (0.81 Mya) (Supplementary Fig. S4-6). All but one of the introgression

events of the Sb supergene variant from *S. invicta/macdonaghi* into the other species occurred in the first quarter of its existence.”

The early divergence between Sb haplotypes does make it possible that degeneration occurred at different rates in different species, and that Sb/Sb queens became completely absent in one species but not the other after the introgression event.

As an aside, Sb males of *S. invicta/macdonaghi* have lower reproductive fitness than SB males in the invasive North American population (Lawson et al. 2012, Proc. R. Soc. B). However, the existence of young Sb/Sb workers, virgin non-reproductive queens, and occasional diploid males (diploid males develop from eggs that are homozygous in the sex-determination locus, a common occurrence in inbred populations) indicates that Sb males can be sufficiently fertile to reproduce. Indeed, such individuals must have been the offspring of a Sb male mating with a heterozygous Sb/SB or homozygous Sb/Sb queen. We suspect that there may be variability in the deleteriousness of Sb across South American populations. We do not discuss this in the manuscript because of the anecdotal nature of the argument.

Finally, it would unfortunately be extremely challenging to experimentally study the viability of Sb homozygote queens and workers. This is because it is currently impossible to perform crosses in the lab, given the difficulty in simulating the mating flight conditions required by fire ants, or performing artificial inseminations (both approaches have often been tried given that fire ants are important agricultural pests in the USA).

Reviewer 2.4 *In summary, there is an important and credible finding here: introgression of a supergene haplotype is likely in this system, calling into question recent conclusions about the evolutionary history of this supergene system. However, the authors don't provide convincing evidence about the direction of introgression; this would require additional analysis and may be limited by the small sample size of the outgroup species that have the SB and Sb genetic polymorphism.*

>>> Authors 2.4: We thank the reviewer for the thoughtful comments.

As we discuss in “Authors 2.2”, we disagree that our original results were ambiguous relative to the direction of the introgression. The location of the SB samples as a sister clade to the *S. invicta/macdonaghi* SB samples in Fig 1e supports an origin of Sb in *S. invicta/macdonaghi* and not in any other species, regardless of the internal position of the samples in the Sb clade. In fact, the main challenge in this paper was to understand whether the SB samples of *S. invicta/macdonaghi* form a sister clade with the Sb samples (the introgression hypothesis) or with the SB samples of the other species (following the ancestral polymorphism hypothesis). We discuss this at length in “Authors 1.1”.

Furthermore, as explained in “Authors 1.1” and “Authors 2.2”, we added four new analyses to robustly test whether the tree topology in Fig. 1e is correct. All four analyses give support to our original topology, *i.e.*, the origin of Sb and SB from an ancestral chromosome in *S. invicta/macdonaghi*. We hope the reviewer agrees that our additional analyses make this interpretation very highly robust, and that the changes in our text make it clear that the *S. AdRX* Sb sample is unlikely to be the sister clade of Sb.

Reviewer 2.5: Additional comments:

1) I found the initial question posed in the introduction (“Is it possible for a complex trait involving alleles at many loci to evolve in one species and then introgress into another?”) to be overly naive in light of our understanding of other sex chromosome/supergene systems. In fact we know the answer to this question based on work from butterfly supergenes and sex chromosomes, for example (some relevant studies are cited at the very end of the discussion, but I think they should be properly acknowledged at the outset, rather than treating this like a completely novel question). ** I would add a sex chromosome example, e.g. 10.1093/molbev/msy181 (there are other examples).

>>> **Authors 2.5:** We understand the reviewer’s point. However, we believe that the initial question frames the area of study of our work in a succinct way, necessary for the broad readership of Nature Communications. We have reworded the end of the first paragraph to ensure we cite previous work at the start of the manuscript as well as in the Discussion section. We also added the citation recommended by the reviewer (line 18):

“Recent studies have shown that supergene variants can spread, unhindered by recombination, not only among populations but also across species barriers^{6–10}.”

2) Lines 140-142: *The authors suggest that adaptive benefits of Sb outweigh the costs of hybridization. In my opinion, this possibility is complicated by the selfish nature of Sb, with both green beard and meiotic drive contributing to its persistence in S. invicta. The sentence implies that it’s adaptive at the organismal level, but I don’t think we know this.*

>>> **Authors:** We appreciate the comment. It is important to recognize that the multiple-queen social form is very likely an adaptive trait. It has evolved more than a dozen times independently across the ants (Boomsma et al. 2014, Animal Behaviour), and the most highly invasive ant pests all have multiple-queen colonies. Nevertheless, we agree with the reviewer that green beard and meiotic drive dynamics likely helped the Sb supergene variant to proliferate in novel species, and that this important idea should be mentioned in our manuscript. We have added the following sentence, now also quoting the relevant paper on the meiotic drive of the system (line 186):

“Over long timescales, the retention of introgressed Sb supergene variants suggests that the costs of hybridization are outweighed by the combination of adaptive benefits of Sb in some environments, the green beard behavior of Sb-carrying workers¹³ (in which they only accept Sb-carrying queens), and, potentially, meiotic drive³³.”

We hope the reviewer agrees that our added explanation is more appropriate.

3) *Out of curiosity, what is going on with the 153 genes that did not converge in the coalescent-based phylogenies? Are there additional signatures of past hybridization genome wide?*

>>> **Authors:** “Not converged” means that the RAxML search for the best tree was aborted with the RAxML error message “did not converge”. Hence for these genes we lack a phylogenetic tree which could be interpreted. As the reviewer implies, the lack of convergence could result from contradictory phylogenetic signals caused by past hybridization. Indeed, as mentioned in Authors 1.1, incongruence between mitochondrial and nuclear trees indicates that between-species hybridization did occur historically (Supplementary Text S9).

However, additional processes and characteristics could also be responsible for contradictory phylogenetic signals, including incomplete lineage sorting, relatively low genetic diversity, and parallel mutations and convergent (or parallel) selection. Some of the resulting patterns could be exacerbated by strong purifying selection on single-copy genes. We suspect that a combination of such mechanisms is responsible for the absence of phylogenetic convergence for the 153 genes.

Overall, we feel that disentangling the reasons for lack of consistent signal at particular loci outside the supergene region is beyond the scope of this manuscript.

4) Line 242: *ABBA-BABBA should be ABBA-BABA*

>>> **Authors:** Thank you — we have corrected this typo now.

5) *It would be interesting to see the source of the samples in the trees, since sampling localities are widely distributed and include native and invasive ranges. Could this be added in a supplementary figure?*

>>> **Authors:** This is a good point. We did include a figure indicating the sampling locations in South America (Supplementary Figure S1):

This figure does illustrate quite clearly the wide and sympatric ranges of some of the species. For example, *S. invicta/macdonaghi* and *S. richteri* are sympatric in Argentina and Uruguay; *S. invicta/macdonaghi*, *S. megergates* and *S. saevissima* are sympatric near Curitiba in Brazil).

We initially tried to include geographic distribution patterns in the interpretation of our data. However, it became apparent that there is geographical admixture within each species (*i.e.*, sub-clades in the species and mitochondrial trees do not cluster geographically). Circumstantial observations suggest that in the very large floodplains in South America, colonies are frequently wiped out or transported vast distances on floating plants and by forming floating “rafts” themselves. This can facilitate rapid dispersal followed by mixture wherever colonies of different types are washed ashore. We ourselves made a circumstantial observation of this type of colony dispersal during sampling, at the mouth of the Paraná River / La Plata River, where a Nature Reserve near Buenos Aires was infested with fire ants after a large flood.

In the end, understanding these patterns is beyond the remit of this manuscript, as they are not directly related to the main question we are addressing, and would not change our conclusions.

Reviewer #3 (Remarks to the Author):

*In this manuscript, the authors use a large-scale genomic dataset to infer the evolutionary history of the social supergene of the *Solenopsis* ants. By comparing a genome-wide phylogeny with a phylogeny made only with genes from the supergene region, they convincingly show that the supergene variant *Sb* was introgressed multiple times between the species of the clade.*

*The biological system studied here is very attractive, and the results described by the authors in this simple study are interesting as they support the fact that inversions can act as “adaptive cassette”, allowing combinations of coadapted alleles to be introgressed as a single unit, and therefore to be protected from recombination. Moreover, this study raises interesting questions on the evolution of supergenes and inversion polymorphisms. The analyses seem to have been done correctly and the literature is properly referenced (but additional references to other supergenes with introgressed alleles are necessary. e.g. white-throated sparrow supergene, *Papilio* supergenes, etc.)*

Overall, I found the manuscript very well written and I appreciated the effort made by the authors to provide a simple manuscript supporting an interesting conclusion. Nevertheless, I found this study lacks details and analyses to properly support the conclusions and assess their validity. This is especially true as this study contradicts recent finding from another team (Yan et al., 2020). Hence, I think that performing additional analyses (detailed bellow) would strongly benefit to this work by providing more support to its conclusion. If these additional analyses are concordant with the analyses provided in this manuscript (which I do not doubt), I think this work will be significant to the field and be of general interest.

>>> Authors: We would like to thank Reviewer 3 for their supportive and kind words. Similarly, the constructive feedback on the analyses raised many valid points that we hope to have addressed in our additional analyses and re-writing of the text.

-Major and Minor comments-

Major:

Reviewer 3.1: *In this manuscript, the authors consider the supergene region as a single genomic unit. However, previous studies have shown that it includes **three chromosomal inversions**. Moreover, Yan et al., 2020, have shown that there are likely recombinations between *SB* and *Sb* haplotypes. This means that different regions of the supergene might have different evolutionary histories. Therefore, instead of providing a single phylogeny for the whole supergene, I would find very interesting and informative to observe how the phylogenies vary along the genome and especially along the supergene. For instance, by computing the support for the different topologies along the genome (e.g. using *Twisst*), as done in Jay et al., *Current Biol.*, 2018, Edelman et al., *Science*, 2019 or Fontaine et al., *Science*, 2015, for example. This could help to understand the discrepancy between Fig 1 and Fig S4, and with the analyses by Yan et al. 2020.*

>>> Authors 3.1: As the reviewer says, the supergene is composed of three inverted regions, each possibly with its own evolutionary history. We thank the reviewer for the suggestion of using phylogenetic weighting (*Twisst*) to understand possible variation in phylogenetic history across the supergene. Because *Twisst* requires multiple samples per species, for this analysis we used samples from *S. invicta/macdonaghi* and *S. richteri* only (as well as outgroup species with no introgression).

- In the large inversion (In(16)1), the introgression topology was the most common, supported by 49% of its windows (18 out of 37 windows). In comparison, the second most common topology, representing a hypothesized ancestral origin of the system, was supported by approximately half as many windows (10 of the 37 windows, 27%).

- In the second inversion In(16)2, both windows with unambiguous support for any particular topology supported the introgression hypothesis.
- The third inversion (In(16)3), near the centromere, included none of the single-copy genes that were used in our analyses (our analyses focus on single-copy genes because they are less likely to be affected by SNP calling artifacts caused by collapsed repeats — which we know are more common in Sb than SB (Stolle et al. 2019; Mol Biol Evol) and which are particularly abundant near the centromere). The fact that all three inversions are found in *S. invicta*, *S. richteri* and *S. AdRX* (Van et al. 2020) makes it very likely that In(16)3 has the same evolutionary history as the other two inversions.

We detail our new topology weighting analysis in the Supplement (Supplementary Text S7) and mention in the main manuscript (line 162):

“Finally, we used a phylogenetic weighting method (Twisst²⁸; Supplementary Text S7) to quantify the support for alternative topologies in non-overlapping windows of four genes across the supergene. Introgression was the most highly supported topology, found in 2.5 times more windows than the topology in which the supergene is an ancestral trans-species polymorphism (Supplementary Fig. S13-14).”

Reviewer 3.2: *The authors seem to consider that Sb evolved first in S. invicta and was then introgressed in other species. I do not see from which analysis this conclusion is derived. I guess this is because Fig. 1 show that the Sb males from other species are nested within the S. invicta Sb males. This should be mentioned in the manuscript.*

>> Authors 3.2: Our conclusion that Sb originated in *S. invicta/macdonaghi* was derived from the fact that all Sb samples cluster as the sister clade to the SB samples of *S. invicta/macdonaghi*, regardless of the internal distribution of samples within the Sb clade. A sentence in the main text explained this key point, but the reviewer’s comment indicates that its wording may not have been sufficiently clear. We have modified this sentence to (line 72):

“Instead, the grouping of Sb males as sister to the *S. invicta/macdonaghi* SB clade shows that the Sb supergene variant originated in chromosome 16 of *S. invicta/macdonaghi* after the divergence between this species and *S. richteri*. The presence of Sb sequences from other species within the *S. invicta/macdonaghi* Sb clade shows that Sb introgressed from *S. invicta/macdonaghi* into the other species (scenario in Fig. 1d)”

Moreover, I think this is not that clear. For instance, Fig 1 seem to show that the S. AdRX Sb male is paraphyletic with other males from the Sb clade. This could mean that the Sb allele could have arisen in S. AdRX and then introgressed to S. invicta. In addition, since Sb and SB do recombine, the position of the samples within the phylogeny could reflect the effect of these recombinations and not only of the introgression events (as suggested by Yan et al. 2020). For instance, the Sb haplotype could have arisen in S. richteri and then introgressed to S. invicta. Recombination between Sb and SB in S. invicta could create this pattern of S. richteri males nested in S. invicta. Additional analyses are required to support the direction of introgression and this should in particular involve sliding window phylogeny or similar analyses (as I suggested before)

>>> Authors 3.2: Reviewer 2 raised similar concerns. We respond to the possibility that Sb originated in *S. AdRX* in Authors 2.2. Briefly, our original analysis, as well as further analyses, support the topology of Sb originating from the clade *S. invicta* SB samples — in other words, Sb seems to have originated from inversions on chromosome 16 of *S. invicta/macdonaghi*.

Furthermore, our original and additional analyses suggest that the Sb of *S. AdRX* is part of an early diverging lineage of Sb, but is not in itself the earliest diverging lineage (as explained in Authors 2.2). A similar pattern is seen for *S. richteri*. We now explicitly mention this in the main manuscript, as explained in Authors 2.2.

We respond to concerns that recombination may be masking the real phylogenetic signal in Authors 2.1. Briefly, recombination between SB and Sb would increase the similarity between SB and Sb in each species, and would either create a tree where SB and Sb cluster by species, or at least lower the support for the node that separates SB and Sb. This is not what we observe.

The reviewer suggests that we should perform additional analysis to test whether the direction of introgression is supported. We thank the reviewer for suggesting the use of topology weighting with Twisst on sliding windows. As we detail above (in Authors 2.2 and Authors 3.1), we have performed four additional analyses: a new dated phylogenetic analysis, the use of QuiBL to differentiate introgression from incomplete lineage sorting, topology weighting with Twisst to understand how support for alternate scenarios varies across the supergene, and an ABBA-BABA-like analysis to explicitly test the origin of introgression. All of these new analyses provided results in line with our original analyses, thus further supporting the introgression hypothesis of Sb originating in an ancestral chromosome in *S. invicta/ macdonaghi*.

We hope that the reviewer agrees that our additional analyses give an overwhelming support for our original interpretation, and that our changes to the main manuscript go a long way to clarify our conclusions.

Reviewer 3.3: *The topology of the S. richteri clade in the whole-genome phylogeny (Fig 1) is very odd. In particular, the position of the samples involved in the introgression event 1 is not normal (for instance, the branch length is far from being poisson-distributed). This peculiar pattern is not visible in fig S4. I wonder about the origin of this particular pattern and I think that the analyses proposed previously might shed some light on it.*

>>> Authors 3.3: Indeed, the samples involved in introgression 1 have a peculiar distribution in the species tree (left of Fig. 1e), with many nested branches of single samples (forming a “pectinate”, “caterpillar”, or asymmetric tree). In ASTRAL-III, the coalescent tree reconstruction software we used, one of the algorithms used to resolve polytomies involves computing caterpillar trees (Zhang et al. 2018, BMC Bioinformatics). Indeed, when looking at trees made using samples from different species, such trees may be a sign of artifactual resolution of polytomies (for an example, see Hosner et al. 2017 MBE, doi:10.1093/molbev/msv347). In our case, however, we are looking at samples of the same species, where polytomies are expected due to the exchange of genetic information within a species. In other words, pectinate distributions can be expected within species, and the exact distribution of *S. richteri* samples in the species tree is not particularly important, as long as it forms a monophyletic group.

Nevertheless, the reviewer’s comment does raise an important point: the *S. richteri* samples that are organized into the pectinate arrangement in the species tree (with the exception of two SB samples) form a separate group in ASTRAL supergene tree, yet they cluster with the other *S. richteri* Sb samples in the concatenated-alignment tree (Supplementary Fig. S7). We have re-written a key sentence in the main manuscript, to point out our uncertainty on whether the introgression of Sb into *S. richteri* occurred once or twice (line 141):

“A notable difference between our two approaches is that the Sb samples of *S. richteri* named as group 1 in the coalescent-based tree are placed into one monophyletic group with the remaining *S. richteri* Sb samples in the concatenation-based tree, suggesting that Sb introgressed only once from *S. invicta* into this species.”

Despite this uncertainty on the number of introgression events into *S. richteri*, our main argument — that Sb introgressed from *S. invicta* to other fire ant species including *S. richteri* — is highly robust, particularly given the additional analyses we have performed.

Reviewer 3.4: *It would be very interesting to perform a datation of the phylogenies, which could allow to infer the time of the introgression events (as before, it should not be performed on the whole supergene since different segments of the supergene might have different histories)*

>>> Authors 3.4: We added an additional analysis (Supplementary Text S3) for dating the species tree and the supergene tree. To create dated phylogenies, we used two previously published dated nodes in the *Solenopsis* phylogeny: the node separating *S. geminata* from the remaining species in our study (3.91 million years) and the node separating all the species in our study with a more ancient *Solenopsis* lineage, represented by the species *Solenopsis fugax* (25.2 million years; Moreau and Bell, Evolution 2013, Ward et al. 2014, Syst Entomol). The considerable divergence time between *S. fugax* and the species in our study means that our SNP calling approach (based on reads mapped to the *S. invicta* reference genome) was inappropriate for this analysis. Instead, we created multiple-sequence alignments from 2,161 coding-sequences retrieved from 22 genome assemblies representing the different species in our study, including 15 that we assembled *de novo*. Compared to the dated phylogeny published by Yan et al. (2020), ours includes two calibration nodes rather than just one, and 2,161 gene sequences rather than just five. While performing this analysis, we also noticed that Yan et al. (2020) assumed that the divergence between *S. geminata* and the other *Solenopsis* species is 4.5 million years, despite citing a study that estimated that divergence between their lineages at 3.91 million years (see Supplementary Text S3).

Our dated species tree provided the backbone (ages of major nodes) to date our existing species (chr1-15) and supergene trees. The results of this analysis are summarized in the main manuscript (line 95):

“A divergence time inference (Supplementary Text S3) suggests that the separation of SB and Sb occurred 0.97 million years ago (Mya), after the split between *S. richteri* and *S. invicta/macdonaghi* (1.01 Mya) and before the diversification of *S. invicta/macdonaghi* (0.81 Mya) (Supplementary Fig. S4-6). All but one of the introgression events of the Sb supergene variant from *S. invicta/macdonaghi* into the other species occurred in the first quarter of its existence. The remaining introgression event is one of the two introgressions into *S. megergates* (numbered 3 in Fig. 1e); its age is within the range measured among individuals in local populations (Supplementary Text S3).”

Reviewer 3.5: *From my point of view, the title of the paper is problematic. A supergene is a locus. A locus cannot be introgressed, only an allele can. Therefore, I think a correct title should be : “Recurring adaptive introgression of a supergene ALLELE that determines social organization” or “Recurring adaptive introgression AT a supergene that determines social organization”. This issue is present in several part of the manuscript. For instance, in “**These supergenes can spread among populations unhindered by recombination, but the extent to which they introgress across species boundaries remains unresolved.**” or in “**The social supergene evolved in one species and subsequently introgressed into others**”. The authors do not talk about the introgression of*

inversions but this is exactly what it is (in the two previous sentences, changing “supergene” to “inversion” would be perfectly correct). The author said that only the Sb haplotypes have been introgressed and not the SB haplotypes, so the whole supergene has not been introgressed. Literally, the supergene have not evolved a single time. The inversions associated with the Sb variant have evolved a single time and have been recurrently involved in supergene formation following their introgression (because they were maintained polymorphic). For me, this study show the recurrent introgression of three inversions associated with a peculiar social organization, which led to the recurrent formation of a supergene underlying polymorphic social organization. This resemble to the situation modeled by Jay et al. BioRxiv 2020. I think it is very important to be very clear on what is a supergene, and for me, a supergene is a set a tightly linked loci segregating together due to recombination suppression. Therefore, I do not think that an inversion is a supergene, it is only an allele of a supergene. I believe talking about the recurrent formation of a supergene following introgression of same inversions would make this study of even more general interest. It raises questions about the force maintaining these inversions polymorphic, which is very interesting considering the ubiquity of inversions in genomes and their recurrent association with complex adaptation.

>>> Authors 3.5: We agree with the reviewer’s definition of a supergene. As the reviewer says, every time Sb introgressed into a new species, it formed a “new” supergene in that species. We agree that this is the appropriate way of thinking about supergene introgression. Unfortunately, we lack the space in this manuscript format to discuss this. However, we do also see the need to adjust the title in light of the reviewer’s comments. We believe the following is appropriate:

“Recurring adaptive introgression of a supergene variant that determines social organization”.

We have concluded that the term “supergene variant” is the most appropriate for describing grouping together the differences between SB and Sb as entire regions (and we have been consistently using this term across many papers). This is because the term “allele” typically is used to refer to localized variant sites (such as single nucleotide polymorphisms). Similarly, the term haplotype is more appropriate for smaller particular regions within the supergene region. We do need the word “supergene” in the title simply because it is an essential keyword. We also changed other sentences in the manuscript: :

“Recent studies have shown that supergene variants can spread, unhindered by recombination, not only among populations but also across species barriers.” (line 18)

And:

“The social supergene variant Sb evolved in one species and subsequently introgressed into others” (line 61)

Minor:

Reviewer 3.6: *Many species are considered as “outgroups”, which is non-sense since some of them are involved in introgression events. I think only the most divergent species (S. geminata) should be considered as “outgroup”, it would make the manuscript more clear.*

>>> Authors 3.6: We agree that this use of the term “outgroups” was ambiguous. In line with the reviewer’s helpful feedback, we have amended the manuscript to refer to the other species used in the analysis in a more general manner, reserving the term “outgroup” for the most divergent *S. geminata* sample.

Reviewer 3.7: *I think the Fig 1e is hard to read. The colors of S. invicta and the outgroups are too similar. The sample labels are useless since we cannot read them. I am not sure it is very useful to have background colors for the part of the figure showing the links between the two phylogenies,*

since it hampers to properly see the links. In addition, for some samples, the link goes all the way to the sample labels (*Outgroup* and *S. richteri*), whereas for others it is limited to the middle part of the figure (*S. invicta*). **This makes the figure look a bit strange. I'm not sure it will improve the figure, but can you try with the same link length for all samples ?**

>>> Authors 3.7: We are grateful for the feedback on our figure. Indeed, the extra lines highlighting introgressed individuals made the figure look inconsistent. To address this, we have contained the links for all individuals to the central panel and used highlighted text rather than extra lines to emphasize the introgressed samples. We believe that the revised figure is easier to understand, while retaining a high amount of information.

Reviewer 3.8: *“Once hybridization occurred, we hypothesize that the type of “green-beard” phenotype encoded by Sb facilitated the acceptance of hybrid Sb-carrying queens into established multiple-queen colonies of S. invicta/macdonaghi.” This sentence is hard to follow and I suggest it could be rephrased. I do not think that referring to the “green-beard” hypothesis is necessary here, as this theory is still controversial and referring to it adds unnecessary complexity*

>>> Authors 3.8: We agree with the reviewer that the sentence was hard to follow. We have modified this sentence and removed the reference to green beard in this instance.

However, we are not aware of any controversy in considering Sb a green beard allele: there is no doubt in the literature that Sb-carrying workers accept only Sb-carrying queens and kill non-carrying queens (e.g., Keller and Ross 1998, Nature and follow-up papers). Of course, it is likely that this is not enough to produce a balanced polymorphism, especially when we consider the differences in life history between the two colony types. In fact, Reviewer 2.5.2 commented that we *should* mention how the green beard effect and meiotic drive in Sb may have had an important role in the spread of this variant in new species. In response, we write further down in the Discussion section (line 186):

“Over long timescales, the retention of introgressed Sb supergene variants suggests that the costs of hybridization are outweighed by the combination of adaptive benefits of Sb in some environments, the green beard behavior of Sb-carrying workers¹³ (in which they only accept Sb-carrying queens), and, potentially, meiotic drive³³.”

Reviewer 3.9: *“Our finding complements recently documented introgressions of smaller supergenes encoding simpler phenotypes (12, 29–31) in other systems.”. Beside the terminology issue raised in (5), I think the use of “simpler” is abusive here, as it implies arbitrary considerations of complexity. For instance, Dixon et al., MBE, 2019 have shown that an introgression has led to the formation of a neo sex chromosome, and sex clearly cannot be considered as a “simpler” phenotype.*

>>> Authors 3.9: The reviewer is of course right that the use of the expression “simpler phenotypes” in relation to other supergenes is at best inaccurate. We have removed that word from the sentence.

However, we would like to note that there is an important difference in the nature of the introgression of Sb in fire ants and that of the introgression of a Y chromosome. Sex determining regions typically act as a “switch” between long-established female and male developmental pathways. The evolution or introgression of a Y chromosome changes the nature of this developmental switch (so to speak), but it does not introduce sex to a previously asexual species. In contrast, the introgression of Sb introduced the multiple-queen social form into species that had single-queen colonies obligatorily. (Of course, Y chromosome introgression may also introduce male-specific phenotypes that evolved originally in other species).

The best-studied examples of introgression are otherwise in butterflies. One of the appealing factors of studying butterfly coloration and patterning is that such phenotypes are generally quite tractable, *i.e.*, they have a relatively direct association between alleles and phenotype. The evolution of multiple-queen colony organization seems more complex, not least because

it involves changes to social behavior. For example, it requires the inhibition of the supernumerary queen killing behavior that is innate in most ant species, the introduction of a system for the recognition of Sb queens by workers, and the emergence of the “budding” dispersal behavior, in which workers and queens separate as a group from their original colony and set a new colony up at a certain distance.

Overall, we do agree with the reviewer that without a better understanding of the genetic bases of these phenotypes, the word “simpler” implies arbitrary considerations of complexity. Nevertheless, we hope the reviewer understands why we think the introgression of the Sb variant is considerably different from previously published examples of introgression.

Reviewer 3.10: *“included an average of at least 10 single-nucleotide differences relative to the reference genome.”*

Not very clear for me what “an average of at least” means. Can you rephrase this sentence ?

>>> Authors 3.10: We apologize for the ambiguous wording. Most single-copy (BUSCO) genes had no or very few genetic variants among the samples in our study, and many of these variants were singletons (present in only one individual). These genes are uninformative for the phylogenetic reconstruction for recently diverged lineages and species. Indeed, the individual gene trees that we originally built from these uninformative genes formed mostly single large polytomies. In consequence, we limited our analysis to those genes that have more signal (among the *S. invicta/macdonaghi* and *S. richteri*) samples. Per gene, we counted the number of SNPs in each individual sample and averaged this number for those samples belonging to *S. invicta/macdonaghi* and *S. richteri*. We modified the sentence in the methods (line 276):

“We retained the most informative 1,881 out of 5,851 genes for which the 342 males identified as *S. invicta*, *S. macdonaghi* and *S. richteri* (the expected closely related ingroup species in the phylogenetic tree of *Solenopsis*) had a per-gene average of at least 10 single-nucleotide differences relative to the reference genome (this step removed non-informative genes that had few or mostly singleton SNPs or that were polymorphic only in the basally diverging species)”

Reviewer 3.11: *“we used ASTRAL-III v5.14.3 (19) to create two coalescent-based trees supported 100 bootstraps”. Not very clear. You created two coalescent-based trees with 100 bootstraps ?*

>>> Authors 3.11: There was a missing word in this sentence which should read: “supported **by** 100 bootstraps”. The sentence was still a bit ambiguous, so we have changed it to (line 288) (line 287):

“we used ASTRAL-III v5.14.324 to create two coalescent-based trees, each supported by 100 bootstraps”

We apologize for the confusion.

To explain ASTRAL’s process in somewhat more detail: For each gene, we had 100 bootstrap RAXML trees. ASTRAL was run 100 times and each time used a different tree from the set of 100 bootstrap trees of each gene. This provides ASTRAL with a bootstrap support value for each node. As stated by the authors of ASTRAL, this Bootstrap support is somewhat less reliable than the local posterior probability derived from quartet scores.

Reviewer 3.12: L31: *“it is possible that its emergence (12) or spread (8, 11, 13)”. As I mentioned in my comment (5), for me, emergence and spread are here exactly the same thing: the introgression of inversions which were subsequently maintained polymorphic.*

>>> Authors 3.12: We agree with the reviewer that this sentence is not particularly clear.

We apologize for the ambiguity and thank the reviewer for their particular attention to the wording. In this sentence, we were simply alluding to a possible **first** “emergence” of Sb:

some supergenes originate through the hybridization of two species with a chromosome segment that is inverted between them (in this case, the “supergene region” is polymorphic in the host species despite not being polymorphic in the original donor species). This has happened, for instance, in the wing pattern supergene in the butterfly *Heliconius numata* (Jay et al. 2018). In contrast, the reviewer pointed out in their previous comment (Reviewer 3.5) that every time Sb crosses a species barrier, a supergene again “emerges” in each host species.

We see now that the reader could not have guessed what type of “emergence” we are alluding to in the Introduction without a long explanation, which would probably overcomplicate the Introduction unnecessarily. We removed the word “emerged” and the associated citation. In any case, we now explain the idea that supergenes can originate by the hybridization of two species with opposing chromosomal orientations further down (line 75):

“The presence of Sb sequences from other species within the *S. invicta/macdonaghi* Sb clade shows that Sb introgressed from *S. invicta/macdonaghi* into the other species (scenario in Fig. 1d). Additionally, the hypothesis that the supergene arose through hybridization of two species with opposing orientations⁶ of the supergene-equivalent region on chromosome 16 is unparsimonious, as it would have required the introgression of Sb back into *S. invicta/macdonaghi* soon after its origin. Instead, the differentiation between Sb and SB suggests that the inversions that suppress recombination between Sb and SB^{12,17,18} in *S. invicta/macdonaghi* are conserved across species.”

Reviewer 3.13: *“be robust to tree discordance due to processes including tree reconstruction errors, incomplete lineage sorting and gene conversion”. Tree discordance due incomplete lineage sorting and gene conversion (and recombination) are informative since they reflect the true history of the sequences studied. I do not think that providing a single tree erasing these differences makes sense*

>>> Authors 3.13: We have responded to this point previously (Authors 3.1). We have added several new analyses, including topology weighting analysis with Twisst that supports our conclusions.

Reviewer 3.14: *L128-137: You discuss here why the Sb variant was successful in establishing in other species. I think it could also be very interesting to quickly discuss here why it remains polymorphic, i.e. why the Sb variant did not fix in populations.*

>>> Authors 3.14: We now briefly mention the polymorphism of the system in all species (line 198):

“The accumulation of deleterious mutations in Sb likely creates a fitness cost that contributes to maintaining the system balanced in *S. invicta*. Further studying the dynamics of Sb degeneration will be essential to understand the evolution of the supergene system across all species.”

Reviewers' Comments:

Reviewer #1:

Remarks to the Author:

In the first version of this manuscript, the authors argued that, based on the topology of a coalescent phylogeny, a supergene had been introgressed between fire ant species. I, along with the other reviewers, were concerned that although this scenario was certainly consistent with the data, there was not clear and convincing evidence that this was indeed the case. The authors have taken our criticisms to heart, and here present three additional analyses that are considered more explicit tests for introgression: QuIBL, Twisst, and a derived allele frequency test (here BBBAA/BBABA). All three of these analyses concur in showing support for introgression of the supergene.

It does appear that the authors have made a small graphing error in figure S8 - based on the order of samples I believe the diagonal line should go from bottom right to top left, not bottom left to top right. However, that does not effect the interpretation of the introgression probabilities which do show increased probability of introgression. My one additional request for this section is that I believe the authors are reporting the genome-wide introgression probability, which is what QuIBL reports by default. However, it would also be important to know the particular introgression probability of windows within the supergene - basically, where does the supergene fall in Figure S9? One would hope that it would be squarely in the second distribution.

Overall, I think the authors have done a lot of work, and a good job of providing necessary, additional support for their hypothesis.

Reviewer #2:

Remarks to the Author:

The authors have added multiple new lines of evidence to provide stronger support for the scenario of Sb introgression laid out in the article. The revised version lays out these arguments more clearly as well. I remain intrigued about the differences in topology across the supergene (as revealed in the TWISST analysis, shown in Figure S13), but delving into the evolutionary scenarios that could lead to this variation along the length of the supergene can be explored in a future study. The authors provided thorough responses to my concerns, and I am satisfied with the revised article.

Reviewer #3:

Remarks to the Author:

Stolle, Pracana and colleagues have addressed all my previous concerns and comments with their latest revision and responses. The resulting manuscript is well improved. Except the very minor points raised below, I have no additional concerns and fully support the publication of this really interesting manuscript

Minor comments:

- 1) « This finding highlights how supergene architecture can enable a complex adaptive phenotype to recurrently permeate species boundaries. » I think using « structural variants » instead of « supergene architecture » will be better !
- 2) L101 « its age is within the range measured among individuals in local populations ». This is not clear. You compare here the introgression age to the age of samples or to their average coalescence time ?
- 2) L194 « where the resulting lack of recombination is associated with an accumulation of deleterious mutation ». Please provide a reference here.
- 4) L198 « The accumulation of deleterious mutations in Sb likely creates a fitness cost that contributes

to maintaining the system balanced in *S. invicta*. Further studying the dynamics of Sb degeneration will be essential to understand the evolution of the supergene system across all species. » I also think you should refer here to other empirical and theoretical work on other supergene systems that have investigated how deleterious mutation load and supergene degeneration can contribute to the observed polymorphism.

RESPONSE TO REVIEWS

REVIEWERS' COMMENTS

Reviewer #1:

In the first version of this manuscript, the authors argued that, based on the topology of a coalescent phylogeny, a supergene had been introgressed between fire ant species. I, along with the other reviewers, were concerned that although this scenario was certainly consistent with the data, there was not clear and convincing evidence that this was indeed the case. The authors have taken our criticisms to heart, and here present three additional analyses that are considered more explicit tests for introgression: QuIBL, Twisst, and a derived allele frequency test (here BBAA/BBABA). All three of these analyses concur in showing support for introgression of the supergene.

>>> We thank the reviewer for the care taken in reviewing our work. We are grateful for the suggested improvements, which, as the reviewer mentions, offer strong support for the introgression of the supergene.

It does appear that the authors have made a small graphing error in figure S8 - based on the order of samples I believe the diagonal line should go from bottom right to top left, not bottom left to top right. However, that does not effect the interpretation of the introgression probabilities which do show increased probability of introgression.

>>> We have changed the diagonal line in Figure S8 according to the reviewer's comment.

My one additional request for this section is that I believe the authors are reporting the genome-wide introgression probability, which is what QuIBL reports by default. However, it would also be important to know the particular introgression probability of windows within the supergene - basically, where does the supergene fall in Figure S9? One would hope that it would be squarely in the second distribution.

>>> The reviewer refers to the Quantifying Introgression via Branch Lengths (QuIBL) method to distinguish between incomplete lineage sorting (ILS) and introgression based on the distribution of internal branch lengths among genomic intervals for a given three-

sample (i.e., three-taxon) subtree or “triplet”. The input data for a QuIBL analysis are a set of trees.

The QuIBL models cannot consider within-locus recombination (Edelman et al. 2019). Thus, to reduce the probability of analysis windows containing a recombination breakpoint, QuIBL is usually run on trees generated from many small (e.g., 5 kb) windows. For reasons mentioned elsewhere in the manuscript, we built each tree from a single-copy gene as identified through the “BUSCO” method.

When the set of input trees spans the entire genome, QuIBL will report genome-wide introgression probabilities — the reviewer believes that this is the case for the data we present. However, we ran QuIBL analyses exclusively using the non-recombining supergene region of chromosome 16. This should be in accordance with the analysis initially suggested by the reviewer. Although we indicated in our response letter that the QuIBL analyses focused on the supergene region, we now realize that this important detail was insufficiently clear in the main text and in the supplementary materials. We apologize for this.

The introgression probabilities we report are only for the supergene region. In our case, for each triplet, QuIBL compares the internal branch length at a locus to the supergene-wide distribution of branch lengths to classify the locus as introgressed or not. It derives introgression probabilities and shows measures of branch lengths.

Here, we compared introgression probabilities among Sb individuals with those among SB individuals. The QuIBL results generally show introgression fractions close to 0 between populations, and much higher patterns that look like “introgression” fractions within populations. This is as things would be expected in normal species. However, the results highlight high introgression fractions across species for comparisons involving Sb haploid males (Figures S8, S10). This is in line with the other findings of our manuscript.

Figure S9 shows four panels, each of which shows a different type of output from four of the many comparisons performed for Figure S8; we examine triplets that would be informative for evaluating introgression between *S. invicta* and *S. richteri*. Again, the data are based entirely on single-copy genes from the supergene region. Each panel shows the distribution of internal branch lengths for a triplet, with one SB individual as the outgroup and Sb individuals of *S. invicta* and *S. richteri* as closest relatives. Each panel shows two peaks, indicating that the two-distribution model (introgression and ILS) fit the data better compared to the one-distribution model (ILS only). This qualitative assessment is supported quantitatively, with $\Delta\text{BIC} > 10$ providing greater support for the two-distribution than the ILS-only model for these samples (ΔBIC is the difference of Bayesian Information Criterion (BIC) scores). The patterns seen here in Figure S9 also hold for Figure S11.

To help you put these figures into perspective, allow us to highlight two examples. In the publication describing QuIBL (Edelman et al. 2019), Figure S62 illustrates the expected distribution of internal branch lengths when a proportion of loci has a history of introgression. Moreover, the documentation of the quibIR package (<https://htmlpreview.github.io/?https://raw.githubusercontent.com/nbedelman/quibIR/master/tutorial.html>) also offers an example based on simulated data to show how introgression results in a bimodal distribution of internal branch lengths for a given triplet. We believe that our QuIBL results clearly fit the expected distribution of internal branch lengths when loci have introgressed.

Overall, I think the authors have done a lot of work, and a good job of providing necessary, additional support for their hypothesis.

Reviewer #2:

The authors have added multiple new lines of evidence to provide stronger support for the scenario of Sb introgression laid out in the article. The revised version lays out these arguments more clearly as well. I remain intrigued about the differences in topology across the supergene (as revealed in the TWISST analysis, shown in Figure S13), but delving into the evolutionary scenarios that could lead to this variation along the length of the supergene can be explored in a future study. The authors provided thorough responses to my concerns, and I am satisfied with the revised article.

>>> We thank the reviewer for the careful review of our work. We agree that the differences in topology across the supergene are potentially interesting, and we agree that delving into the evolutionary scenarios that may cause this variation is an important next step for a future study.

Reviewer #3:

Stolle, Pracana and colleagues have addressed all my previous concerns and comments with their latest revision and responses. The resulting manuscript is well improved. Except the very minor points raised below, I have no additional concerns and fully support the publication of this really interesting manuscript

>>> We thank the reviewer for the careful review of our work. As we mentioned before, we are very grateful for the suggestions given by all reviewers, which have been key for us to corroborate our initial argument.

Minor comments:

1) *« This finding highlights how supergene architecture can enable a complex adaptive phenotype to recurrently permeate species boundaries. » I think using « structural variants » instead of « supergene architecture » will be better !*

>>> We understand the point raised by the reviewer. However, the suggested change would complicate the message of our short abstract, since at this point of the article we have not yet explained that the supergene of the fire ant is associated with an inversion. An additional, minor point is that supergenes can also exist independently of structural variation (*i.e.*, through other recombination suppression mechanisms, as occurs in some frog sex chromosomes).

2) L101 *« its age is within the range measured among individuals in local populations ». This is not clear. You compare here the introgression age to the age of samples or to their average coalescence time ?*

>>> We agree with the reviewer that this sentence could have been more clear. We have changed it to:

“the age of the node representing this introgression event is within the range of nodes separating individuals of the same species”

2) L194 *« where the resulting lack of recombination is associated with an accumulation of deleterious mutation ». Please provide a reference here.*

>>> We now reference this sentence.

4) L198 *« The accumulation of deleterious mutations in Sb likely creates a fitness cost that contributes to maintaining the system balanced in S. invicta. Further studying the dynamics of Sb degeneration will be essential to understand the evolution of the supergene system across all*

species. » *I also think you should refer here to other empirical and theoretical work on other supergene systems that have investigated how deleterious mutation load and supergene degeneration can contribute to the observed polymorphism.*

>>> We have added the following sentence:

“The accumulation of deleterious mutations in *Sb* likely creates a fitness cost that contributes to maintaining the system balanced in *S. invicta*. Indeed, this type of effect has been predicted by theoretical studies of supergene evolution^{38,39}, and suggested to occur in the supergenes of species like the butterfly *Heliconius numata*⁴⁰, the white-throated sparrow⁴¹, the ruff⁴² and the seaweed fly⁴³.”